# Learning Space Partitions for Path Planning

**Kevin Yang**[1][*]    **Tianjun Zhang**[1][*]    **Chris Cummins**[2]    **Brandon Cui**[2]    **Benoit Steiner**[2]
**Linnan Wang**[3]    **Joseph E. Gonzalez**[1]    **Dan Klein**[1]    **Yuandong Tian**[2]
[1]UC Berkeley    [2]Facebook AI Research    [3]Brown University
{yangk,tianjunz,jegonzal,klein}@berkeley.edu
{cummins,bcui,benoitsteiner,yuandong}@fb.com
linnan_wang@brown.edu

## Abstract

Path planning, the problem of efficiently discovering high-reward trajectories, often requires optimizing a high-dimensional and multimodal reward function. Popular approaches like CEM [37] and CMA-ES [16] greedily focus on promising regions of the search space and may get trapped in local maxima. DOO [31] and VOOT [22] balance exploration and exploitation, but use space partitioning strategies independent of the reward function to be optimized. Recently, LaMCTS [45] empirically learns to partition the search space in a reward-sensitive manner for black-box optimization. In this paper, we develop a novel formal regret analysis for when and why such an adaptive region partitioning scheme works. We also propose a new path planning method LaP$^3$ which improves the function value estimation within each sub-region, and uses a latent representation of the search space. Empirically, LaP$^3$ outperforms existing path planning methods in 2D navigation tasks, especially in the presence of difficult-to-escape local optima, and shows benefits when plugged into the planning components of model-based RL such as PETS [7]. These gains transfer to highly multimodal real-world tasks, where we outperform strong baselines in compiler phase ordering by up to 39% on average across 9 tasks, and in molecular design by up to 0.4 on properties on a 0-1 scale. Code is available at https://github.com/yangkevin2/neurips2021-lap3.

## 1   Introduction

Path planning has been used extensively in many applications, ranging from reinforcement learning [7, 13, 14] and robotics [27, 35, 26] to biology [24], chemistry [40], material design [21], and compiler optimization [42]. The goal is to find the most rewarding trajectory (i.e., state-action sequence) $\mathbf{x} = (s_0, a_0, s_1, \ldots, s_n)$ in the search space $\Omega$: $\mathbf{x}^* = \arg\max_{\mathbf{x} \in \Omega} f(\mathbf{x})$, where $f(\mathbf{x})$ is the reward.

In this work, we focus on deterministic path planning problems with long trajectories $\mathbf{x}$, and discontinuous and/or multimodal reward functions $f$. Such high-dimensional non-convex optimization problems exist in many real domains, both continuous and discrete. While we could always find near-optimal $\mathbf{x}$ by random sampling given an infinite query budget, in practice we prefer a sample-efficient method that achieves high-reward trajectories with fewer queries of the reward function $f$.

While global methods like Bayesian Optimization (BO) [3] may struggle with limited samples and high-dimensional spaces, classic approaches like CEM [37] and CMA-ES [16] learn a local model around promising trajectories. For example, CEM tracks a population of trajectories and repeatedly re-samples its population according to the highest-performing trajectories from the previous generation. On the other hand, such a focus can trap CEM in local optima, as confirmed empirically (Sec. 5).

Other recent approaches, such as VOOT [22] and DOO [31], use a (recursive) region partitioning scheme: they split the search space $\Omega$ into sub-regions $\Omega = \Omega_1 \cup \ldots \cup \Omega_k$, then invest more samples into promising sub-regions while continuing to explore other regions via an *upper confidence bound*

35th Conference on Neural Information Processing Systems (NeurIPS 2021).

(UCB). While such exploration-exploitation procedures adaptively focus on promising sub-regions and lead to sub-linear regret and optimality guarantees, their *region partition* procedure is manually designed by humans and remains non-adaptive. For example, DOO partitions the space with uniform axis-aligned grids and VOOT with Voronoi cells, both independent of the reward $f$ to be optimized.

Recently, Wang et al. proposed LaNAS [46] and LaMCTS [45], which *adaptively* partition the search regions based on sampled function values, and focus on good regions. They achieve strong empirical performance on Neural Architecture Search (NAS) and black-box optimization, outperforming many existing methods including evolutionary algorithms and BO. Notably, in recent NeurIPS'20 black-box optimization challenges, two teams that use variants of LaMCTS ranked 3rd [38] and 8th [23].

In this paper, we provide a simple theoretical analysis of LaMCTS to reveal the underlying principles of adaptive region partitioning, an analysis missing in the original work. Based on this analysis, we propose **La**tent Space **P**artitions for **P**ath **P**lanning (LaP$^3$), a novel optimization technique for path-planning. Unlike LaMCTS, LaP$^3$ uses a latent representation of the search space. Additionally, we use the maximum (instead of the mean) as the node score to improve sample efficiency, verified empirically in Sec. 5.3. Both changes are motivated by our theoretical analysis.

We verify LaP$^3$ on several challenging path-planning tasks, including 2D navigation environments from past work with difficult-to-escape local optima, and real-world planning problems in compiler optimization and molecular design. In all tasks, LaP$^3$ demonstrates substantially stronger exploration ability to escape from local optima compared to several baselines including CEM, CMA-ES and VOOT. On compiler phase ordering, we achieve on average 39% and 31% speedup in execution cycles comparing to -O3 optimization and OpenTuner [1], two widely used optimization techniques in compilers. On molecular design, LaP$^3$ outperforms all of our baselines in generating molecules with high values of desirable properties, beating the best baseline in average property value by up to $0.4$ on properties in a $[0, 1]$ range. Additionally, extensive ablation studies show factors that affect the quality of planning and verify the theoretical analysis.

LaP$^3$ is a general planning technique and can be readily plugged into existing algorithms with path planning components. For example, we apply LaP$^3$ to PETS [7] in model-based RL and observe substantially improved performance for high-dimensional continuous control and navigation, compared to CEM as used in the original PETS framework.

## 2 Latent Space Monte Carlo Tree Search (LaMCTS)

LaMCTS [45] is recently proposed to solve black-box optimization problems $\mathbf{x}^* = \arg\max_{\mathbf{x}} f(\mathbf{x})$ via recursively learning $f$-dependent region partitions. Fig. 1 and Alg. 1 show the details of LaMCTS as well as our proposed approach LaP$^3$ (formally introduced in Sec. 4) for comparison.

---

**Algorithm 1** LaP$^3$ Pseudocode for Path Planning. Improvements over LaMCTS in green.

---

1: **Input:** Number of rounds $T$, Environment Oracle: $f(\mathbf{x})$, Dataset $\mathcal{D}$, Sampling Latent Model $h(\mathbf{x})$, Partitioning Latent Model $s(\mathbf{x})$.
2: **Parameters:** Initial #samples $N_{\text{init}}$, Re-partitioning interval $N_{\text{par}}$, Node partition threshold $N_{\text{thres}}$, UCB parameter $C_p$.
3: Pre-train $h(\cdot)$ on $\mathcal{D}$ when $\mathcal{D} \neq \emptyset$.
4: Set region partition $\mathcal{V}_0 = \{\Omega\}$.
5: Draw $N_{\text{init}}$ samples uniformly from $\mathcal{S}_0 = \{(\mathbf{x}_i, f(\mathbf{x}_i))\}_{i=1}^{N_{\text{init}}} \subset \Omega$.
6: **for** $t = 0, \ldots, T - N_{\text{init}} - 1$ **do**
7:     **if** $t$ divides $N_{\text{par}}$ **then**
8:         Train/fine-tune latent model $h(\cdot)$ using samples $\mathcal{S}_t \cup \mathcal{D}$ (Eqn. **??**).
9:         Re-learn region partition $\mathcal{V}_t \leftarrow \text{Partition}(\Omega, \mathcal{S}_t, N_{\text{thres}}, s(\cdot))$ in latent space $\Phi_s$ of $s(\cdot)$.
10:     **end if**
11:     **for** $k := \text{root}, k \notin \mathcal{V}_{\text{leaf}}$ **do**

12:         $k \leftarrow \arg\max\limits_{\Omega_c \in \text{child}(\Omega_k)} b_c$, where $b_c := \left[ \frac{1}{n(\Omega_c)} \sum_{\mathbf{x}_i \in \Omega_c} f(\mathbf{x}_i) \max\limits_{\mathbf{x}_i \in \Omega_c} f(\mathbf{x}_i) + C_p \sqrt{\frac{2 \log n(\Omega_k)}{n(\Omega_c)}} \right].$

13:     **end for**
14:     Initialize CMA-ES using encodings of $\mathcal{S}_t \cap \Omega_k$ via $h(\cdot)$. Here $\Omega_k$ is the chosen leaf sub-region.
15:     $\mathcal{S}_t \leftarrow \mathcal{S}_{t-1} \cup \{(\mathbf{x}_t, f(\mathbf{x}_t))\}$, where $\mathbf{x}_t$ is drawn from CMA-ES and decoded via $h^{-1}(\cdot)$.
16: **end for**

---

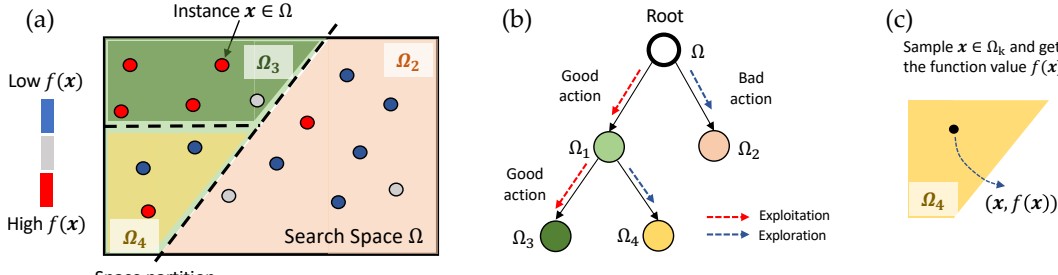

Figure 1: LaP³ extends LaMCTS [45] to path planning. **(a)** Starting from a search space $\Omega$, both LaP³ and LaMCTS first draw a few samples $\mathbf{x} \in \Omega$, then learn to partition $\Omega$ into a sub-region $\Omega_1$ with good samples (high $f(\mathbf{x})$) and a sub-region $\Omega_2$ with bad samples (low $f(\mathbf{x})$). Compared to LaMCTS, LaP³ uses a *latent space* and reduces the dimensionality of the search space. **(b)** Sampling follows the learned recursive space partition, focusing on good regions while still exploring bad regions using UCB. LaP³ uses the *maximum* of the sampled value in a region ($\max_{\mathbf{x}_i \in \Omega} f(\mathbf{x}_i)$) as the node value, while LaMCTS uses the mean. **(c)** Upon reaching a leaf, new data points are sampled within the region and the space partition is relearned.

LaMCTS starts with $N_{\text{init}}$ random samples of the entire search space $\Omega$ (line 5 in Alg. 1). For a region $\Omega_k$, let $n(\Omega_k)$ be the number of samples within. LaMCTS dictates that, if $n(\Omega_k) \geq N_{\text{thres}}$, then $\Omega_k$ is partitioned into disjoint sub-regions $\Omega_k = \Omega_{\text{good}} \cup \Omega_{\text{bad}}$ as its *children* (Fig. 1(a)-(b), line 9 in Alg. 1, the function Partition). Intuitively, $\Omega_{\text{good}}$ contains promising samples with high $f$, while $\Omega_{\text{bad}}$ contains samples with low $f$. Unlike DOO and VOOT, such a partition is learned using $\mathcal{S}_t \cap \Omega_k$, our samples so far in the region, and is thus dependent on the function $f$ to be optimized.

Given tree-structured sub-regions, new samples are mostly drawn from promising regions and occasionally from other regions for exploration. This is achieved by Monte Carlo Tree Search (MCTS) [4] (line 11-13): at each tree branching, the UCB score $b$ is computed to balance exploration and exploitation (line 12). Then the subregion with highest UCB score is selected (e.g., it may have high $f$ and/or low $n$). This is done recursively until a leaf sub-region $\Omega'$ is reached. Then a new sample $\mathbf{x}$ is drawn from $\Omega'$ (line 15) either uniformly, or from a local model constructed by an existing optimizer (e.g., TuRBO [10], CMA-ES [16]), in which case LaMCTS becomes a meta-algorithm. When more samples are collected, regions are further partitioned and the tree gets deeper.

Finally, the function Partition in Alg. 1 is defined as follows: first a 2-class K-means on $(\mathbf{x}, f(\mathbf{x}))$ is used to create positive/negative sample groups. Next, a SVM classifier is used to learn the decision boundary (hence the partition), so that samples with high $f(\mathbf{x})$ fall into $\Omega_{\text{good}}$, and samples with low $f(\mathbf{x})$ fall into $\Omega_{\text{bad}}$ (Fig. 1(a)). See Appendix A for the pseudo code. The partition boundary can also be re-learned after more samples are collected (line 9).

## 3 A Theoretical Understanding of Space Partitioning

While LaMCTS [45] shows strong empirical performance, it contains several components with no clear theoretical justification. Here we attempt to give a formal regret analysis when sub-regions $\{\Omega_k\}$ are *fixed* and all at the same tree level, and the function $f$ is deterministic. We leave further analysis of tree node splitting and evolution of hierarchical structure to future work.

Despite the drastic simplification, our regret bound still shows why an $f$-dependent region partition is helpful. By showing that a better regret bound can be achieved by a clever region partition as empirically used in the Partition function in Alg. 1, we justify the design of LaMCTS. Furthermore, our analysis suggests several empirical improvements over LaMCTS and motivates the design of LaP³, which outperforms multiple classic approaches on hard path planning problems.

### 3.1 Regret Analysis with Fixed Sub-Regions

We consider the following setting. Suppose we have $K$ $d$-dimensional regions $\{\Omega_k\}_{k=1}^{K}$, and $n_t(\Omega_k)$ is the visitation count at iteration $t$. The global optimum $\mathbf{x}^*$ resides in some unknown region $\Omega_{k^*}$. At each iteration $t$, we visit a region $\Omega_k$, sample (uniformly or otherwise) a data point $\mathbf{x}_t \in \Omega_k$, and retrieve its *deterministic* function value $f_t = f(\mathbf{x}_t)$. In each region $\Omega_k$, define $\mathbf{x}_k^* := \arg\max_{\mathbf{x} \in \Omega_k} f(\mathbf{x})$ and the maximal value $g^*(\Omega_k) = f(\mathbf{x}_k^*)$. The maximal value *so far* at iteration $t$ is $g_t(\Omega_k) = \max_{t' \leq t} f(\mathbf{x}_{t'})$. It is clear that $g_t \leq g^*$ and $g_t \to g^*$ when $t \to +\infty$.

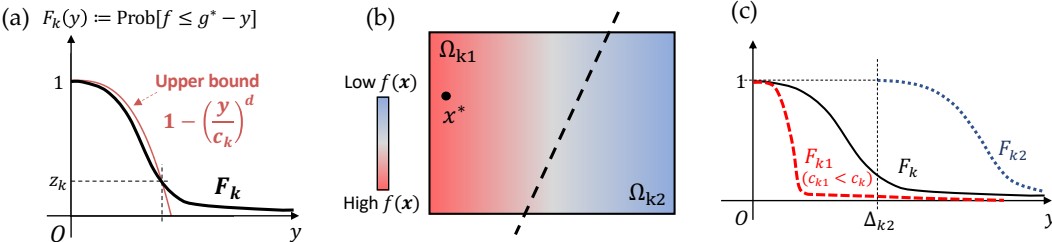

Figure 2: Theoretical understanding of space partitioning. **(a)** Definition of $(z_k, c_k)$-diluted region $\Omega_k$ (Def. 1). **(b)** Partition of region $\Omega_k$ into good region $\Omega_{k1}$ and bad region $\Omega_{k2}$. Optimal solution $\mathbf{x}^* \in \Omega_{k1}$. **(c)** After space partitioning, $F_k$ is split into $F_{k1}$ and $F_{k2}$. The good region $F_{k1}$ has much smaller $c_{k1}$ while the bad region has much larger best-to-optimality gap $\Delta_{k2}$. As a result, the expected total regret decreases.

We define the *confidence bound* $r_t = r_t(\Omega_k)$ so that with high probability, the following holds:

$$g_t(\Omega_k) \geq g^*(\Omega_k) - r_t(\Omega_k) \tag{1}$$

At iteration $t$, we pick region $k_t$ to sample based on the upper confidence bound: $k_t = \arg\max_k g_t(\Omega_k) + r_t(\Omega_k)$. Many different confidence bounds can be applied; for convenience in this analysis, we use the "ground truth" bound from the cumulative density function (CDF) of $f$ within the region $\Omega_k$ (Please check Appendix B for all proofs):

**Lemma 1.** *Let $F_k(y) := \mathbb{P}\left[f(\mathbf{x}) \leq g^*(\Omega_k) - y | \mathbf{x} \in \Omega_k\right]$ be a strictly decreasing function, and let $r_{k,t}(\Omega_k) := F_k^{-1}\left(\delta^{1/n_t(\Omega_k)}\right)$. Then Eqn. 1 holds with probability $1 - \delta$.*

Here $F_k^{-1}$ is the inverse function of $F_k$ and randomness arises from sampling within $\Omega_k$. Since $F_k$ is a strictly decreasing function, $F_k^{-1}$ exists and is also strictly decreasing. By definition, $F_k \in [0, 1]$, $F_k(0) = 1$ and $F_k^{-1}(1) = 0$. We then define the *dilution* of each region as follows:

**Definition 1** (($z_k, c_k$)-dilution). *A region $\Omega_k$ is $(z_k, c_k)$-diluted if there exist $z_k, c_k$ such that $F_k(y) \leq 1 - (y/c_k)^d$ for $y \in [0, c_k(1 - z_k)^{1/d}]$, where $z_k$ is the smallest $F_k(y)$ to make the inequality hold.*

The intuition for dilution for a given region, as depicted in Fig. 2(a), is that all but $z_k$ fraction of the region has function value close to the maximum, with "close" defined based on $c_k$ (smaller $c_k$ implies a stricter definition of "close"). Obviously if $\Omega_k$ is $(z_k, c_k)$-diluted then it is $(z'_k, c'_k)$-diluted for any $c'_k \geq c_k$ and $z'_k \geq z_k$. Therefore, we often look for the smallest $z_k$ and $c_k$ to satisfy the condition. If a region $\Omega_k$ has small $c_k$ and $z_k$, we say it is *highly concentrated*. For example, if $f(\mathbf{x})$ is mostly constant within a region, then $c_k$ is very small since $F_k(y)$ drops to 0 very quickly. In such a case, most of the region's function values are concentrated near the maximum, making it easier to optimize.

While the definition of concentration may be abstract, we show it is implied by Lipschitz continuity:

**Corollary 1.** *If a region $\Omega_k$ is $L_k$-Lipschitz continuous, i.e., $|f(\mathbf{x}) - f(\mathbf{x}')| \leq L_k \|\mathbf{x} - \mathbf{x}'\|_2$, and there exists an $\epsilon_0$-ball $B(\mathbf{x}_k^*, \epsilon_0) \subseteq \Omega_k$, then with uniform sampling, $\Omega_k$ is $(1 - \epsilon_0^d \tilde{V}_k^{-1}, L_k \sqrt[d]{\tilde{V}_k})$-diluted. Here $\tilde{V}_k := V_k/V_0$ is the relative volume with respect to the unit sphere volume $V_0$.*

Typically, a smoother function (with small $L_k$) and large $\epsilon_0$ yield a less diluted (and more concentrated) region. However, the concept of dilution (Def. 1) is much broader. For example, if we shuffle function values within $\Omega_k$, Lipschitz continuity is likely to break but Def. 1 still holds.

Now we will bound the total regret. Let $R_t(a_t) := f^* - g_t(\Omega_{a_t}) \geq 0$ be the regret of picking $\Omega_{a_t}$ and $R(T) := \sum_{t=1}^T R_t(a_t)$ be the total regret, where $T$ is the total number of samples (queries to $f$). Define the *gap* of each region $\Delta_k := f^* - g^*(\Omega_k)$ and split the region indices into $\mathcal{K}_{\text{good}} := \{k : \Delta_k \leq \Delta_0\}$ and $\mathcal{K}_{\text{bad}} := \{k : \Delta_k \geq \Delta_0\}$ by a threshold $\Delta_0$. $C_{\text{good}} := \left(\sum_{k \in \mathcal{K}_{\text{good}}} c_k^d\right)^{1/d}$ and $C_{\text{bad}} := \left(\sum_{k \in \mathcal{K}_{\text{bad}}} c_k^d\right)^{1/d}$ are the $\ell_d$-norms of the $c_k$ in these two sets. Finally, $M := \sup_{\mathbf{x} \in \Omega} f(\mathbf{x}) - \inf_{\mathbf{x} \in \Omega} f(\mathbf{x})$ is the maximal gap between function values. Treating each region $\Omega_k$ as an arm and applying a regret analysis similar to multi-arm bandits [41], we obtain the following theorem:

**Theorem 1.** *Suppose all $\{\Omega_k\}$ are $(z_k, c_k)$-diluted with $z_k \leq \eta/T^3$ for some $\eta > 0$. The total expected regret $\mathbb{E}[R(T)] = \mathcal{O}\left[C_{\text{good}} \sqrt[d]{T^{d-1} \ln T} + M(C_{\text{bad}}/\Delta_0)^d \ln T + KM\eta/T\right]$.*

## 3.2 Implications of Theorem 1

**The effect of space partitioning**. Reducing $\{c_k\}$ results in a smaller regret $R(T)$. Thus if we can partition $\Omega_k$ into two sub-regions $\Omega_{k1}$ and $\Omega_{k2}$ such that the good partition $\Omega_{k1}$ has smaller $c_{k1} < c_k$ and the bad partition $\Omega_{k2}$ has larger $\Delta_{k2} > \Delta_0$ and falls into $\mathcal{K}_{\mathrm{bad}}$, then we can improve the regret bound (Fig. 2(b)-(c)). This coincides with the $\mathrm{Partition}$ function of LaMCTS very well: it samples a few points in $\Omega_k$, and trains a classifier to separate high $f$ from low $f$. On the other hand, if we partition a region $\Omega_k$ randomly, e.g., each $f(\mathbf{x})$ is assigned to either $\Omega_{k1}$ or $\Omega_{k2}$ at random, then statistically $F_{k1} = F_{k2} = F_k$ and $c_{k1} = c_{k2} = c_k$, which *increases* the regret bound. Therefore, the partition needs to be *informed* by data that have already been sampled within the region $\Omega_k$.

**Recursive region partitioning**. In Theorem 1, we assume all regions $\{\Omega_k\}$ have fixed $c_k$ and $z_k$, so the bound breaks for large enough $T$ (as $\eta/T^3$ eventually becomes smaller than any fixed $z_k$). However, as LaMCTS conducts further internal partitioning within $\Omega_k$, its $c_k$ and $z_k$ keep shrinking with more samples $T$. If each split leads to slightly fewer bad $f$ (i.e., lighter "tail"), with the ratio being $\gamma < 1$, then by the definition of CDF, $z_k$ is the probability mass of the tail and thus $z_k \sim \gamma^{-T/N_{\mathrm{par}}}$. This would yield $z_k \leq \eta/T^3$ for all $T$, since $\gamma^{-T}$ decays faster than $1/T^3$ and Theorem 1 would hold for all $T$. See Appendix F.2 for empirical verification of decaying $z_k$.

## 3.3 Related Work and Limitations

While related to Lipschitz bandits [28] and coarse-to-fine deterministic function optimization like DOO and SOO [32], our analysis is fundamentally different. We have discussed how $f$-dependent region partitioning and a data-driven learning procedure affect the regret bound, which to our knowledge has not been previously addressed. See Appendix B.5 for further remarks on Theorem 1.

There is more work to be done to fully understand how LaMCTS works. In particular, we did not analyze when to split a node (e.g. how many samples we need to collect before making a decision), or the effect of relearning the space partition. We also have not considered stochastic reward functions, where the maximum function value in the sub-region may no longer be the best metric of goodness. We leave these to future work.

# 4 $\mathrm{LaP}^3$ for Path Planning

Based on our analysis, we propose $\mathrm{LaP}^3$, which extends LaMCTS to path planning, a problem with temporal structure. $\mathrm{LaP}^3$ outperforms baseline path planning approaches in both continuous and discrete path planning problems. Here we represent trajectories as action sequences $\mathbf{x} = (a_0, a_1, \dots, a_{n-1})$ and treat them as high-dimensional vectors $\mathbf{x}$ in the trajectory space $\Omega$.

Thus, $\mathrm{LaP}^3$ searches over the space $\Omega$, recursively partitioning $\Omega$ into subregions based on trajectory reward, and sampling from subregions using CMA-ES [16] (which is faster than TuRBO [10] used in the original LaMCTS). We emphasize again that $\mathrm{LaP}^3$'s region partitioning procedure is fully adaptive, in contrast to traditional MCTS approaches such as VOOT, which only partition the trajectory space based on one action at a time.

Additionally, we have made several improvements over the original LaMCTS, as detailed in Algorithm 1. First, we use the maximal value $\max_{i \in \Omega_k} f(\mathbf{x}_i)$ rather than the mean value $\frac{1}{n(\Omega_k)} \sum_{i \in \Omega_k} f(\mathbf{x}_i)$ as the metric of goodness for each node $k$ (and its associated region $\Omega_k$). This is driven by Theorem 1, which gives a regret bound based on maximum values. Intuitively, using the mean value would cause the algorithm to be slow to respond to newly discovered territory: it takes time for the mean metric to boost, and we may miss important leaves. We show the difference empirically in Sec. 5.3.

Second, Theorem 1 suggests that a lower-dimensional (smaller $d$) and smoother (smaller $c_k$) representation leads to lower regret. Therefore, $\mathrm{LaP}^3$ employs a latent space as described below.

## 4.1 Latent Spaces For Partitioning and Sampling

$\mathrm{LaP}^3$ leverages a latent space $\Phi_s$ for the *partition* space, by passing $\Omega$ through some encoder $s$. That is, we disentangle the *sampling* space $\Omega$ from which we sample new candidate trajectories, from the *partition* space $\Phi_s$ on which we construct the search space partition. Critically, we do not need $s^{-1}$: we never decode from $\Phi_s$ back to $\Omega$. Thus $s$ can dramatically reduce the dimension of the partition

space, which may improve regularization due to the small number of samples, without suffering large reconstruction loss. $s$ will be fixed rather than learned in this case. Once the partition has been constructed on $\Phi_s$, and we select a leaf region to propose from, we sample new $\mathbf{x}$ from $\Omega$ as before.[1]

In principle, the sampling space can itself be a latent space $\Phi_h$, with an encoder $h$ and decoder $h^{-1}$. That is, one runs the inner solver in $\Phi_h$ to propose samples before decoding back to $\Omega$. $h$ could be a principal component analysis (PCA) [49], a random network encoding [43], or a reversible flow [9], depending on the environment's particular $\Omega$ and state/action structure. While some latent representations can be fixed by specifying the inductive bias (e.g., random network encoding), others can be learned from data, optimizing reconstruction loss $\min_h \mathbb{E}_{\mathbf{x}} \left[ w(\mathbf{x}) \| h^{-1}(h(\mathbf{x})) - \mathbf{x} \|^2 \right]$, where $w(\mathbf{x})$ is a weighting function emphasizing trajectories with high cumulative reward $f(\mathbf{x})$. In this case, $h$ and $h^{-1}$ may be fine-tuned using each new $(\mathbf{x}, f(\mathbf{x}))$ pair when $\texttt{LaP}^3$ proposes and queries a new trajectory $\mathbf{x}$, or they may be pre-trained using a set of unlabeled $\mathbf{x}$ with $w(\mathbf{x}) \equiv 1$. For consistency in our main experiments, we do not use a latent $\Phi_h$, although we observe that using this second latent space can yield a slight performance in some environments (Appendix F.7).

# 5    $\texttt{LaP}^3$ on Synthetic Environments

We test $\texttt{LaP}^3$ on a diverse set of environments to evaluate its performance in different settings.

**Baselines**. We compare $\texttt{LaP}^3$ to several baselines. **LaMCTS** is the original LaMCTS algorithm using CMA-ES as an inner solver, like $\texttt{LaP}^3$. **Random Shooting (RS)** [36] samples random trajectories and returns the best one. **Cross-Entropy Methods (CEM)** [2] use the top-$k$ samples to fit a local model to guide future sampling. A related approach, **Covariance matrix adaptation evolution strategy (CMA-ES)** [16], tracks additional variables for improved local model fitting. **Voronoi optimistic optimization applied to trees (VOOT)** [22] is a "traditional" MCTS method for continuous action spaces that builds a tree on actions at each timestep. **iLQR** [26] is a seminal gradient-based local optimization approach used extensively in controls. Finally, **proximal policy optimization (PPO)** [39] is a standard reinforcement learning algorithm.

$\texttt{LaP}^3$ does not require substantially more tuning effort than CEM or CMA-ES, the best-performing among our baselines experimentally. The only additional hyperparameter tuned in $\texttt{LaP}^3$ is the $C_p$ controlling exploration when selecting regions to sample from, which is dependent on the scale of the reward function. However, our $C_p$ only varies by a factor of up to 10 across our diverse environments, and performance is not overly sensitive to small changes (Appendix F.5).

We use MiniWorld [5] for continuous path planning and MiniGrid [6] for discrete.

## 5.1    MiniWorld

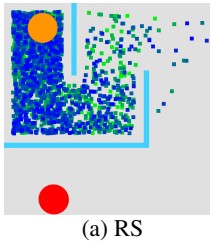
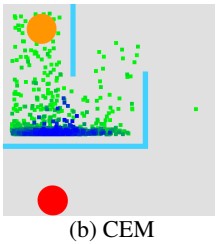
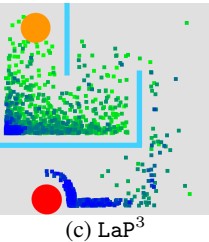

(a) RS                    (b) CEM                    (c) $\texttt{LaP}^3$

Figure 3: MazeS3 environment. **Start:** Orange circle. **Goal:** Red circle. Dots indicate final agent positions of 2,000 proposed trajectories (green: first iteration, blue: last iteration). CEM gets stuck in a local optimum of reward (shown as concentration of blue dots), while $\texttt{LaP}^3$ succeeds in reaching the goal.

We consider the following 2D navigation tasks in *MiniWorld*. **MazeS3**: Agent navigates in a 3 by 3 maze to a goal. Greedy path planning gets stuck in local optima (Figure 3). **FourRooms**: Agent navigates from one room in a 2 by 2 configuration to a goal in the diagonally opposite room. Greedy path planning gets stuck in a corner. **SelectObj**: Open space with two goals. Large final reward when

---

[1]Specifically, we initialize the inner solver (CMA-ES in our experiments) using the pre-existing samples corresponding to the selected leaf region in $\Phi_s$, and then propose new samples using that initialization.

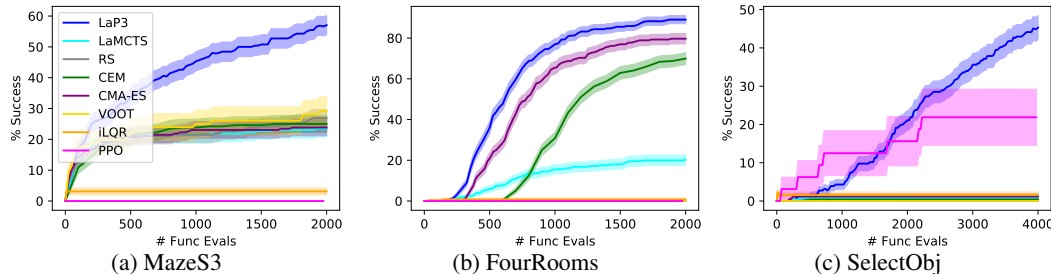

(a) MazeS3  (b) FourRooms  (c) SelectObj

Figure 4: Mean, and standard deviation of mean (256 trials; fewer for VOOT and PPO due to speed), of success rate across MiniWorld tasks. LaP³ significantly outperforms all baselines on all three tasks.

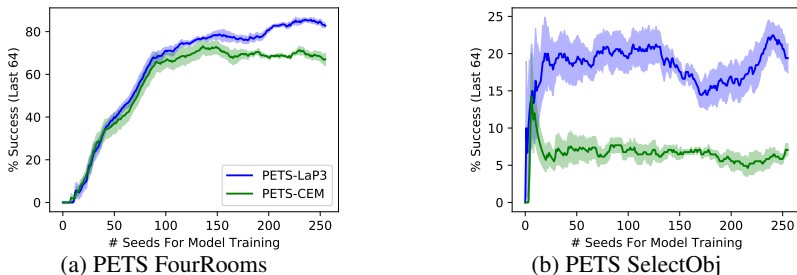

(a) PETS FourRooms  (b) PETS SelectObj

Figure 5: LaP³ in PETS compared to original PETS planners on MiniWorld environments, using PETS-learned world models. Sliding length-64 window of success percentage against number of training seeds for world model. LaP³ significantly outperforms all baselines on both tasks.

reaching the farther goal, while a distance-based reward misleadingly points to the closer goal. For full environment specifics, see Appendix H.1.

We modify the original setup to use a continuous action space ($\Delta x$ and $\Delta y$), and provide a sparse reward (proximity to goal, with an additional bonus for reaching the goal) at end-of-episode. We use a high-dimensional top-down image view as the state. We featurize this image using a randomly initialized convolutional neural network, a reasonable feature extractor as shown in [43]. LaP³ uses periodic snapshots of the featurized state as the partition space $\Phi_s$. That is, we collect all the observed states over the course of the full trajectory, and then form the latent space by concatenating every $n^{th}$ state (here $n = 20$), while discarding the rest to reduce overall dimensionality. Success is defined using a binary indicator for reaching the goal (far goal for SelectObj).

**Results**. LaP³ substantially outperforms all baselines on all three tasks, despite heavily tuning the baselines' hyperparameters (Appendix G), showing that LaP³ works for challenging tasks containing suboptimal local maxima. In MazeS3, LaP³ succeeds but CEM gets stuck (Figure 3). VOOT, which builds an MCTS tree on actions at each timestep, struggles on all environments; LaP³ can be viewed as an extension of MCTS that performs better on such long-horizon tasks. PPO also performs poorly, perhaps due to the sparse reward given only at the end of an episode, and the relatively small (for RL) number of episodes. In the most difficult SelectObj task, LaP³ solves nearly half of environment seeds within 4,000 queries of the oracle, whereas most baselines—including the original LaMCTS—quickly reach the near goal but struggle to escape this local optimum.

We also evaluate LaP³ when combined with a model-based approach, PETS [7], on FourRooms and SelectObj (omitting MazeS3 because the changing maze walls for each seed make it difficult to learn a world model). Following PETS' setting and due to difficulty in learning image-based world models [12, 14], we use 2D agent position as the state. As shown in Fig. 5, LaP³ substantially outperforms the authors' original CEM implementation in the PETS framework, demonstrating that it is not reliant on access to the oracle model but can work with learned models as well.

## 5.2  MiniGrid

*MiniGrid* [6] is a popular sparse-reward symbolic environment for benchmarking RL algorithms. It contains tasks with discrete states and actions such as **DoorKey (DK)**: pick up a key and open

the door connecting two rooms; **MultiRoom (MR)**: traverse several rooms by opening doors; and **KeyCorridor (KC)**, a combination of MR and DK: some doors are locked and require a key. As in MiniWorld, we add proximity to the goal to the final sparse reward.

In discrete action spaces, $\text{LaP}^3$ optimizes the vector of all action probabilities over all timesteps, and takes the highest-probability action at each step. As in MiniWorld, we use periodic state snapshots featurized by a randomly initialized CNN as the partition space $\Phi_s$. We compare $\text{LaP}^3$ to the same baselines as in MiniWorld, except VOOT and iLQR which are designed for continuous tasks.

**Results**. $\text{LaP}^3$ is equal to or better than baselines on all six tasks (Table 1). Especially in the hardest tasks with the most rooms (MR-N4S5, MR-N6), $\text{LaP}^3$ improves substantially over baselines.

|  | **DK-6** | **DK-8** | **KC-S3R3** | **KC-S3R4** | **MR-N4S5** | **MR-N6** |
|---|---|---|---|---|---|---|
| LaMCTS | **0.96±0.02** | 0.09 ± 0.17 | -2.63±0.09 | -4.43±0.13 | -14.71±0.87 | -118.70±4.68 |
| RS | **0.97±0.01** | **0.34±0.13** | -2.38±0.09 | **-4.27±0.12** | -18.16±0.80 | -119.39±4.64 |
| CEM | 0.03±0.12 | -3.34±0.34 | -3.40±0.08 | -4.93±0.13 | -22.88±1.00 | -131.32±5.24 |
| CMA-ES | 0.93±0.03 | 0.23±0.14 | -2.46±0.09 | -4.44±0.12 | -14.31±0.78 | **-117.50±4.61** |
| $\text{LaP}^3$ | **0.95±0.03** | **0.46±0.13** | **-2.27±0.09** | **-4.37±0.13** | **-11.68±0.75** | **-113.53±4.49** |

Table 1: Results for $\text{LaP}^3$ in MiniGrid. $\text{LaP}^3$ is equal or better on all tasks (higher is better).

### 5.3 Analysis

We run several ablations on $\text{LaP}^3$ in MiniWorld to justify our methodological choices. See Appendix F for further analysis on hyperparameter sensitivity, UCB metric, and latent spaces.

**Region Selection in $\text{LaP}^3$**. We consider four alternative region selection methods. (1) $\text{LaP}^3$-mean: using mean function value rather than max for UCB, as in LaMCTS [45]; **(2)** $\text{LaP}^3$-nolatent: not using a latent space for partitioning; **(3)** $\text{LaP}^3$-notree: directly selecting the leaf with the highest UCB score; and **(4)** $\text{LaP}^3$-noUCB: only using node value rather than UCB. $\text{LaP}^3$ greatly outperforms all variations in MiniWorld, justifying our design.

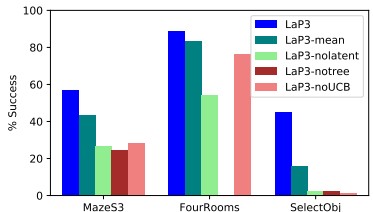

Figure 6: MiniWorld success percentages with different region selection methods.

|  | **MazeS3** | **FourRooms** | **SelectObj** |
|---|---|---|---|
| $L_k$ | 87.5 | 100.0 | 100.0 |
| $c_k$ | 81.3 | 93.8 | 100.0 |

Table 2: Percentage out of 32 environment seeds on MiniWorld environments where $\text{LaP}^3$ yields a better estimated Lipschitz and $c_k$ compared to random partitioning on the same nodes.

**Data-driven space partition in $\text{LaP}^3$ vs. random partitioning**. We examine $c_k$ in Def. 1 and Lipschitz constant $L_k$ in Corollary 1 to verify the theory. We conduct a preliminary analysis on $\text{LaP}^3$'s tree after the full 2,000 queries (4,000 for SelectObj). At each intermediate node, we estimate $L_k$ and $c_k$ of its children from the $\text{LaP}^3$ partition, against a random partition that divides the node's samples with the same ratio (see Appendix F.1 for estimation details). We then average the values for both $\text{LaP}^3$ and random partitions over all nodes in the tree. We find that $\text{LaP}^3$ does yield lower average $L_k$ and $c_k$ (Table 2), indicating that our data-driven space partition is effective.

## 6 $\text{LaP}^3$ on Real-World Applications

### 6.1 Compiler Phase Ordering

Compiler optimization applies a series of program transformations from a set of predefined optimizations (e.g., *loop invariant code motion*, *function inlining* [30]) to improve code performance. Since these optimizations are not commutative, the order in which they are applied is extremely important. This problem, known as *phase ordering*, is a core challenge in the compiler community. Current

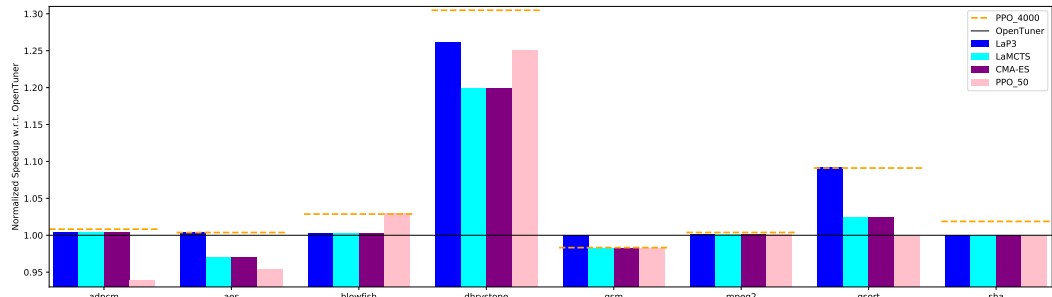

Figure 7: Compiler phase ordering results, in terms of normalized execution cycles with respect to Open-Tuner [1], a widely used method for program autotuning. LaP³ is consistently equal or better compared to baselines. We omitted the matmul task since it doesn't fit the scale with its 245% speedup over OpenTuner.

solutions to this NP-hard problem rely heavily on heuristics: groups of optimizations are often packed into "optimization levels" (such as -O3 or -O0) hand-picked by developers [34, 42].

We apply LaP³ to the standard CHStone benchmarks [17], and use periodic snapshots of states as $\Phi_s$ and the identity as $\Phi_h$. See Appendix H.2 for full environment details.

**Results**. LaP³ is 31% faster on average compared to OpenTuner, and 39% compared to -O3 (not shown in figure). Compared to a stronger PPO baseline using 50 samples (PPO_50) and to CMA-ES, we achieve up to 10% and 7% speedup respectively. Finally, compared to final PPO results at convergence after 4000 samples (PPO_4000) as an oracle, LaP³ does similarly on most tasks, despite being much more sample efficient (only 50 samples). Full results in Appendix E.

## 6.2 Molecular Design

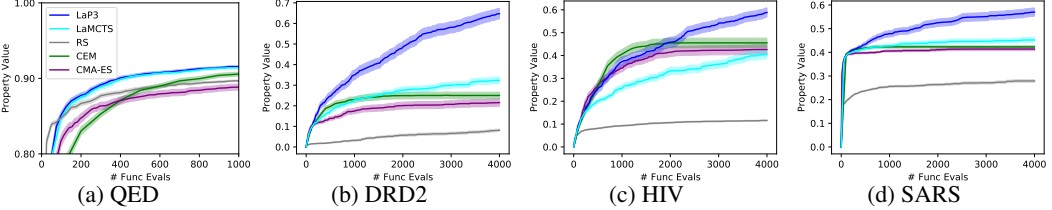

Figure 8: Mean and standard deviation (128 trials), of max property value discovered in molecular design tasks. LaP³ significantly outperforms all baselines on all properties.

Finally, we evaluate LaP³ on molecular design. Given an oracle for a desired molecular property, the goal is to generate molecules with high property score after the fewest trials. This is critical to pharmaceutical drug development [44], as property evaluations require expensive wet-lab assays.

Similar to [18], we fix a query budget and optimize several properties: **QED**: a synthetic measure of drug-likeness, relatively simpler to optimize; **DRD2**: a measure of binding affinity to a human dopamine receptor; **HIV**, the probability of inhibition potential for HIV; and **SARS**: the same probability for a variant of the SARS virus, related to the SARS-CoV-2 virus responsible for COVID-19. All four properties have a range of $[0, 1]$; higher is better. For DRD2, HIV, and SARS, we evaluate using computational predictors from [33] (DRD2) and [51] (HIV, SARS) in lieu of wet-lab assays.

To run LaP³ on molecular design, we view the molecular string representation (SMILES string [48]) as the action sequence, similar to how many generative models generate molecules autoregressively [11, 25, 8, 50]. Following the state-of-the-art HierG2G model from [19], we learn a latent representation from a subset of ChEMBL [29], a dataset of 1.8 million drug-like molecules, *without* using any of its property labels (e.g., effectiveness in binding to a particular receptor). During this unsupervised training, we only use the 500k molecules with the lowest property scores to ensure a good molecule is discovered by search rather than a simple retrieval from the dataset. Our setting differs from many existing methods for molecular design, which assume a large preexisting set of molecules with the desired property for training the generator [33, 20, 52, 50].

On this task only, the latent space is trained on additional unlabeled data, and is used as both the partition space $\Phi_s$ and sampling space $\Phi_h$ for LaP$^3$. All baselines operate in the same space for fair comparison. Otherwise, all methods struggle to generate well-formed molecules of reasonable length.

**Results**. Figure 8 shows the highest property score discovered by each method for each property. The absolute difference is small in the relatively simple synthetic QED task. However, LaP$^3$ outperforms all baselines by a much greater margin—up to 0.4 in DRD2—in the more challenging and realistic DRD2, HIV, and SARS tasks, where CEM and CMA-ES quickly plateau but LaP$^3$ continues to improve with more function evaluations.

## 7 Conclusion

We propose LaP$^3$, a novel meta-algorithm for path planning that learns to partition the search space so that subsequent sampling focuses more on promising regions. We provide a formal regret analysis of region partitioning, motivating improvements that yield large empirical gains. LaP$^3$ particularly excels in environments with many difficult-to-escape local optima, substantially outperforming strong baselines on 2D navigation tasks as well as real-world compiler optimization and molecular design.

## Acknowledgments and Disclosure of Funding

We thank the members of the Berkeley NLP group as well as our four anonymous reviewers for their helpful feedback. This work was supported by Berkeley AI Research, and the NSF through a fellowship to the first author.

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
