# A LaMCTS Partition Function

Algorithm 2 details the pseudocode for the partition function used in LaMCTS, which we use in $\texttt{LaP}^3$ as well.

---

**Algorithm 2** Partition Function

---

1: **Input:** Input Space $\Omega$, Samples $\mathcal{S}_t$, Node partition threshold $N_{\text{thres}}$, Partitioning Latent Model $s(\mathbf{x})$
2: Set $\mathcal{V}_0 = \{\Omega\}$
3: Set $\mathcal{V}_{queue} = \{\Omega\}$
4: **while** $\mathcal{V}_{queue} \neq \emptyset$ **do**
5:     $\Omega_p \leftarrow \mathcal{V}_{queue}.pop(0)$
6:     **if** $n(\Omega_p) \geq N_{\text{thres}}$ **then**
7:         $S_{good}, S_{bad} \leftarrow$ samples from $S_t$ corresponding to indices of $k$-means$(s(\Omega_p \cap S_t))$
8:         Fit SVM on $S_{good}, S_{bad}$
9:         Use SVM to split $\Omega_p$ into $\Omega_{good}, \Omega_{bad}$
10:        $\mathcal{V}_0 \leftarrow \mathcal{V}_0 \cup \{\Omega_{good}, \Omega_{bad}\}$
11:        $\mathcal{V}_{queue} \leftarrow \mathcal{V}_{queue} \cup \{\Omega_{good}, \Omega_{bad}\}$
12:     **end if**
13: **end while**
14: **return** $\mathcal{V}_0$

---

# B Proofs

## B.1 Proof of Lemma 1

*Proof.* Let $\delta < 1$. Define the following cumulative density function (CDF):

$$F_k(y) := \mathbb{P}[f(\mathbf{x}) \leq g_k^* - y | \mathbf{x} \in \Omega_k] \tag{2}$$

where $g_k^* := \sup_{\mathbf{x} \in \Omega_k} f(\mathbf{x})$. It is clear that $F_k(y)$ is a monotonically decreasing function with $F_k(0) = 1$ and $\lim_{y \to +\infty} F_k(y) = 0$. Here we assume it is strictly decreasing so that $F_k(y)$ has a well-defined inverse function $F_k^{-1}$.

In the following, we will omit the subscript $k$ for brevity. Let us bound $\mathbb{P}[g_t \geq g^* - y]$:

$$\begin{align} \mathbb{P}[g_t \geq g^* - y] &= 1 - \mathbb{P}[g_t \leq g^* - y] \tag{3} \\ &\overset{\text{①}}{=} 1 - \prod_i \mathbb{P}[f(\mathbf{x}_i) \leq g^* - y | \mathbf{x}_i \in \Omega_k] \tag{4} \\ &= 1 - F_k^{n_t}(y) \tag{5} \end{align}$$

Note that ① is due to the fact that all samples $\mathbf{x}_1, \ldots, \mathbf{x}_{n_t}$ are independently drawn within the region $\Omega_k$. Given $\delta$, let $r_t := F_k^{-1}(\delta^{1/n_t})$ and we have:

$$\mathbb{P}[g_t \geq g^* - r_t] = 1 - \delta \tag{6}$$

$\square$

## B.2 Proof of Corollary 1

*Proof.* Since $f$ is $L_k$-Lipschitz over region $\Omega_k$, we have:

$$|f(\mathbf{x}) - f(\mathbf{x}')| \leq L_k \|\mathbf{x} - \mathbf{x}'\|_2 \quad \forall \mathbf{x}, \mathbf{x}' \in \Omega_k \tag{7}$$

Since the optimal solution $\mathbf{x}_k^* \in \Omega_k$ is in the interior of $\Omega_k$, there exists $\epsilon_0$ so that $B(\mathbf{x}_k^*, \epsilon_0) \subseteq \Omega_k$. From the Lipschitz condition, we know that in the ball $B(\mathbf{x}_k^*, \epsilon)$ with $\epsilon \leq \epsilon_0$, the function values are also quite good:

$$f(\mathbf{x}) \geq f(\mathbf{x}_k^*) - L_k \|\mathbf{x} - \mathbf{x}_k^*\|_2 = g^* - L_k \epsilon, \quad \forall \mathbf{x} \in B(\mathbf{x}_k^*, \epsilon) \tag{8}$$

Therefore, at least in the ball of $B(\mathbf{x}_k^*, \epsilon)$, all function values are larger than a threshold $g^* - L_k\epsilon$. This means that for $\epsilon \leq \epsilon_0$:

$$F_k(L_k\epsilon) = \mathbb{P}\left[f(\mathbf{x}) \leq g^* - L_k\epsilon | \mathbf{x} \in \Omega_k\right] \leq 1 - \frac{V_0\epsilon^d}{V_k} \tag{9}$$

where $V_0$ is the volume of the unit $d$-dimensional sphere. Letting $\tilde{V}_k := V_k/V_0$ be the relative volume with respect to unit sphere, we have:

$$F_k(y) \leq 1 - \frac{(y/L_k)^d}{\tilde{V}_k} = 1 - \left(\frac{y}{L_k\tilde{V}_k^{1/d}}\right)^d \quad \text{when } y \leq L_k\epsilon_0 \tag{10}$$

Therefore, $\Omega_k$ is at most $(1 - \epsilon_0^d\tilde{V}_k^{-1}, L_k\tilde{V}_k^{1/d})$-diluted with $z_k = 1 - \epsilon_0^d\tilde{V}_k^{-1}$ and $c_k = L_k\tilde{V}_k^{1/d}$. □

### B.3 New Lemma and Proof

**Lemma 2.** *If $\Omega_k$ are $(z_k, c_k)$-diluted, then for any $\delta \in [z_k, 1]$ and $j \geq 1$, we have:*

$$F_k^{-1}(\delta^{1/j}) \leq c_k \sqrt[d]{\frac{1}{j}\ln\frac{1}{\delta}} \tag{11}$$

*Proof.* Note that the diluted condition $F_k(y) \leq 1 - \left(\frac{y}{c_k}\right)^d$ for $y \in [0, c_k\sqrt[d]{1 - z_k}]$ can be also be written as:

$$F_k^{-1}(z) \leq c_k\sqrt[d]{1 - z}, \quad \forall z \in [z_k, 1] \tag{12}$$

Since now we have $z_k \leq \delta \leq \delta^{1/j} \leq 1$ for any $j \geq 1$, following Eqn. 12 we have:

$$F_k^{-1}(\delta^{1/j}) \leq c_k\sqrt[d]{1 - \delta^{1/j}} \tag{13}$$

Due to the inequality that for $a < 1$ and $x > 0$, $a^x \geq 1 + x\ln a$ (which can be proven by simply showing the derivative is non-negative), if we take $a = \delta$ and $x = 1/j$, we have:

$$\delta^{1/j} \geq 1 - \frac{1}{j}\ln\frac{1}{\delta} \tag{14}$$

which gives:

$$F_k^{-1}(\delta^{1/j}) \leq c_k\sqrt[d]{j^{-1}\ln 1/\delta} \tag{15}$$

□

### B.4 Proof of Theorem 1

*Proof.* Take $\delta = \eta/T^3$ so that $\delta \geq z_k$ for all regions $\Omega_k$. Then Eqn. 1 holds for all $T$ iterations and all $K$ arms with probability at least $1 - KT\delta$ by union bound, which we consider a "good event".

For brevity, define $g_k^* := g(\Omega_k)$ as the optimal function value within the region of $\Omega_k$ and $n_{k,t} := n_t(\Omega_k)$ the visitation count of region $\Omega_k$. Define $\Delta_k := f^* - g_k^*$ the minimal regret of each arm and $r_{k,t} := r_t(\Omega_k)$ the confidence bound. At iteration $t$, since we pick $k = a_t$ as the region to explore, it must be the case that:

$$f^* + r_{k,t} \overset{\text{①}}{\geq} g_k^* + r_{k,t} \overset{\text{②}}{\geq} g_{k,t} + r_{k,t} \overset{\text{③}}{\geq} g_{k^*,t} + r_{k^*,t} \overset{\text{④}}{\geq} g_{k^*}^* = f^* \tag{16}$$

where $k^*$ is the index of the optimal region $\Omega_{k^*}$ where its maximum $g_{k^*}^*$ is the global optimal value $f^*$. Here ① is due to global optimality of $f^*$, ② is due to global optimality of $g_k^*$ within region $\Omega_k$: $g_k^* \geq g_{k,t}$, ③ is due to the fact that we pick $a_t = k$ at iteration $t$, and ④ is due to the non-negativity of the confidence bound: $r_{k^*,t} \geq 0$. Therefore, since $g_k^* + r_{k,t} \geq f^*$, we have:

$$\Delta_k := f^* - g_k^* \leq r_{k,t} \tag{17}$$

Now we bound the total regret.

Note that for $R_t(a_t) := f^* - g_{a_t,t}$, we have:

$$R_t(a_t) := f^* - g_{a_t,t} = f^* - g_{a_t}^* + g_{a_t}^* - g_{a_t,t} \leq 2r_{a_t,t} \tag{18}$$

due to the fact that $\Delta_k = f^* - g_k^* \leq r_{k,t}$ and the property of the confidence bound that $g_{k,t} \geq g_k^* - r_{k,t}$ with $k = a_t$.

One the other hand, using Lemma 2, we also have

$$\Delta_k \leq r_{k,t} = F_k^{-1}(\delta^{1/n_{k,t}}) \leq c_k \sqrt[d]{\frac{1}{n_{k,t}} \ln \frac{1}{\delta}} \tag{19}$$

which means that

$$n_{k,t} \leq \left(\frac{c_k}{\Delta_k}\right)^d \ln \frac{1}{\delta} \tag{20}$$

So if the region $\Omega_k$ has a large gap $\Delta_k$, then $n_{k,t}$ would have a small upper-bound (and be small). As a result, we would never visit that region after a fixed number of visitations. This also helps bound the regret.

If we sum over $R_t(a_t)$ over $t$ iterations, we get $R(T)$. We could reorganize them into two kinds of regions, the good regions where $\mathcal{K}_{\text{good}} := \{k : \Delta_k \leq \Delta_0\}$ and the bad regions where $\mathcal{K}_{\text{bad}} := \{k : \Delta_k > \Delta_0\}$:

$$R(T) = \sum_{t=1}^{T} R_t(a_t) = \underbrace{\sum_{a_t \in \mathcal{K}_{\text{good}}} R_t(a_t)}_{R_{\text{good}}(T)} + \underbrace{\sum_{a_t \in \mathcal{K}_{\text{bad}}} R_t(a_t)}_{R_{\text{bad}}(T)} \tag{21}$$

Let $M := \sup_{\mathbf{x} \in \Omega} f(\mathbf{x}) - \inf_{\mathbf{x} \in \Omega} f(\mathbf{x})$ be the maximal gap between the highest and lowest function values. Note that $M$ is also the largest regret for a single move at any iteration. Letting $C_{\text{bad}} := \left(\sum_{k \in \mathcal{K}_{\text{bad}}} c_k^d\right)^{1/d}$ be the $\ell_d$-norm of $c_k$ over bad regions, we then have:

$$R_{\text{bad}}(T) \leq M \left(\frac{C_{\text{bad}}}{\Delta_0}\right)^d \ln \frac{1}{\delta} \tag{22}$$

$$R_{\text{good}}(T) = 2 \sum_{k \in \mathcal{K}_{\text{good}}} \sum_{j=1}^{n_{k,T}} r_{k,t}\Big|_{n_{k,t}=j} = 2 \sum_{k \in \mathcal{K}_{\text{good}}} \sum_{j=1}^{n_{k,T}} F_k^{-1}(\delta^{1/j}) \tag{23}$$

For $R_{\text{good}}(T)$, this is because for each region $k$ we visit it $n_{k,T}$ times and each time we pay a price that is proportional to $1/n_{k,t}$ for $n_{k,t} = 1 \ldots n_{k,T}$.

Using Lemma 2, since all $\Omega_k$ are $(z_k, c_k)$-concentrated and $z_k \leq \delta$, this leads to:

$$R_{\text{good}}(T) \leq 2 \sqrt[d]{\ln 1/\delta} \sum_{k \in \mathcal{K}_{\text{good}}} c_k \sum_{j=1}^{n_{k,T}} j^{-1/d} \tag{24}$$

Assuming $d > 1$ (high-dimensional case), we use the bound

$$\sum_{j=1}^{n} j^{-1/d} \leq \frac{d}{d-1} n^{1-1/d} \tag{25}$$

and we have:

$$R_{\text{good}}(T) \leq \frac{2d}{d-1} \sqrt[d]{\ln 1/\delta} \sum_{k \in \mathcal{K}_{\text{good}}} c_k n_{k,T}^{\frac{d-1}{d}} \tag{26}$$

Hölder's inequality says if $1/p + 1/q = 1$, then $\sum_k |x_k y_k| \leq (\sum_k |x_k|^p)^{1/p} (\sum_k |y_k|^q)^{1/q}$. Using it with $p = d$ and $q = \frac{d}{d-1}$, we get

$$R_{\text{good}}(T) \leq \frac{2d}{d-1} \sqrt[d]{\ln \frac{1}{\delta}} \left(\sum_{k \in \mathcal{K}_{\text{good}}} c_k^d\right)^{\frac{1}{d}} \left(\sum_{k \in \mathcal{K}_{\text{good}}} n_{k,T}\right)^{\frac{d-1}{d}} \tag{27}$$

$$\leq \frac{2d}{d-1} \sqrt[d]{\ln \frac{1}{\delta}} C_{\text{good}} T^{\frac{d-1}{d}} \tag{28}$$

where $C_{\text{good}} := \left( \sum_{k \in \mathcal{K}_{\text{good}}} c_k^d \right)^{\frac{1}{d}}$ is the $\ell_d$-norm of $c_k$ over good regions.

Finally, if a good event doesn't happen (with probability $KT\delta$), we would pay a regret of at most $M$ at each iteration $t$, yield a bound of $MKT^2\delta$ for $T$ iterations.

Since $\delta = \eta/T^3$ then finally we have

$$\mathbb{E}\left[R(T)\right] = \mathcal{O}\left[ C_{\text{good}} \sqrt[d]{T^{d-1} \ln T} + M \left( \frac{C_{\text{bad}}}{\Delta_0} \right)^d \ln T + KM\eta/T \right] \tag{29}$$

$\square$

### B.5 Additional Implications of Theorem 1

**Relationship w.r.t sample complexity.** Note that one can turn the regret bound of $R(T)$ in Theorem 1 into sample complexity: if there exists $T$ such that $\mathbb{E}\left[R(T)\right]/T \leq \epsilon$, then with high probability there exists at least one $R_t(a_t) := f^* - g_t(\Omega_{a_t}) \leq \epsilon$, showing that we already found a good $\mathbf{x} \in \Omega_{a_t}$ with $f(\mathbf{x}) = g_t(\Omega_{a_t}) \geq f^* - \epsilon$. To achieve this, since $R(T) \sim T^{\frac{d-1}{d}}$, we set $R(T)/T \sim T^{-\frac{1}{d}}$. Then the sample complexity $T$ to achieve the *global optimum* within an $\epsilon$-ball is $\sim 1/\epsilon^d$, which is the best we can achieve without structured information on $f$. Previous papers [41] show a slightly worse bound $\mathcal{O}(T^{\frac{d+1}{d+2}})$ since they also consider stochastic functions and discretization error.

**Which region to split?** Since $C_{\text{good}} := \left( \sum_{k \in \mathcal{K}_{\text{good}}} c_k \right)^{1/d}$ is an $\ell_d$-norm, when $d$ is large (i.e., high-dimensional), $C_{\text{good}} \sim \max_{k \in \Omega_{\text{good}}} c_k$ so ideally we should split the region with the highest $c_k$ to reduce $C_{\text{good}}$ the most. Intuitively this means the most diluted / scattered region.

## C Model-Based Reinforcement Learning

LaP$^3$ can escape local minima and achieve significantly better results in various RL tasks using a simulated environment. In MiniWorld, we showed that we could also plug LaP$^3$ into the PETS [7] framework, replacing the CEM method that was originally used as a planner (Sec. 5). Here we additionally use Mujoco, a commonly used benchmark, to validate the performance. Note that Mujoco is a very smooth task and doesn't contain many local minima, so traditional methods work reasonably well in this domain. In Tab. 3, we can see that in easier tasks like Reacher and Pusher, LaP$^3$ is a little worse than CEM. However, in hard tasks like Halfcheetah and Walker, LaP$^3$ has over 1000 reward gain over baseline methods.

|  | swimmer | acrobot | hopper | pendulum | halfcheetah |
|---|---|---|---|---|---|
| PETS(RS) | **12.92**±**7.92** | -41.93±2.17 | -1525.39±222.43 | 130.14±28.39 | 497.03±121.72 |
| PETS(CEM) | -6.87±1.30 | -24.77±7.63 | -2102.57±136.35 | 153.05±12.00 | 271.01±165.08 |
| LaP$^3$ | 10.82±8.47 | **6.29**±**10.29** | **-1205.01**±**167.52** | **153.70**±**38.02** | **3942.47**±**400.01** |

|  | reacher | pusher | ant | I-pendulum | walker |
|---|---|---|---|---|---|
| PETS(RS) | -1165.59±12.04 | -220.58±2.94 | 1330.81±113.17 | -11.87±10.43 | -1204.94±344.70 |
| PETS(CEM) | **-36.45**±**2.87** | **-90.70**±**7.29** | **1405.56**±**46.94** | -4.84±5.29 | -2036.28±213.41 |
| LaP$^3$ | -40.31±5.05 | -103.42±2.91 | 1033.46±148.87 | **-0.30**±**0.09** | **-53.25**±**987.53** |

Table 3: Results for Mujoco with replanning frequency of 5. We see that LaP$^3$ performs substantially better than CEM and RS in hard tasks like Halfcheetah and Walker.

## D Evaluation on Synthetic Functions

We additionally evaluate LaP$^3$ on some synthetic functions (Ackley and Levy functions, both 20-dimensional and 100-dimensional) used in the original LaMCTS paper [45]. For these tasks we compare to just the original LaMCTS method. Both LaP$^3$ and LaMCTS use the TuRBO inner solver following [45] for the 20-dimensional version of both functions, and the CMA-ES inner solver for

the 100-dimensional version for computational efficiency. LaP$^3$ performs equal or better on these tasks (Table 4; note lower is better).

| | Ackley-20D | Levy-20D | Ackley-100D | Levy-100D |
|---|---|---|---|---|
| LaMCTS | $0.48 \pm 0.03$ | $0.51 \pm 0.09$ | $0.65 \pm 0.25$ | $14.24 \pm 4.87$ |
| LaP$^3$ | $0.49 \pm 0.04$ | $0.34 \pm 0.07$ | $0.46 \pm 0.15$ | $11.95 \pm 3.56$ |

Table 4: LaP$^3$ vs the original LaMCTS method on some synthetic functions evaluated in the original LaMCTS work. Note *lower* is better. LaP$^3$ performs equal or better on these tasks.

# E  Tables of Numerical Results

We provide in Tables 5 through 9 the numerical final rewards for our tasks, corresponding to the plots in the main text.

| | MazeS3 | FourRooms | SelectObj |
|---|---|---|---|
| LaMCTS | $23.4 \pm 2.6$ | $20.3 \pm 2.5$ | $0.8 \pm 0.6$ |
| RS | $0.4 \pm 0.4$ | $0.0 \pm 0.0$ | $3.1 \pm 1.1$ |
| CMA-ES | $23.8 \pm 2.7$ | $79.7 \pm 2.5$ | $1.2 \pm 0.7$ |
| CEM | $25.0 \pm 2.7$ | $69.9 \pm 2.9$ | $0.4 \pm 0.4$ |
| VOOT | $26.2 \pm 2.7$ | $0.0 \pm 0.0$ | $0.0 \pm 0.0$ |
| RandDOOT | $25.0 \pm 2.7$ | $0.0 \pm 0.0$ | $0.0 \pm 0.0$ |
| iLQR | $3.1 \pm 1.1$ | $0.8 \pm 0.6$ | $1.6 \pm 0.8$ |
| PPO | $0.0 \pm 0.0$ | $0.0 \pm 0.0$ | $31.3 \pm 8.2$ |
| LaP$^3$ | $\mathbf{57.0 \pm 3.1}$ | $\mathbf{89.1 \pm 2.0}$ | $\mathbf{45.3 \pm 3.1}$ |

Table 5: Results (success percentage over 256 trials) for MiniWorld tasks for different methods, querying oracle transition model. LaP$^3$ substantially outperforms all baselines on all three environments. This table corresponds to Figure 4.

| | FourRooms | SelectObj |
|---|---|---|
| PETS-RS | $0.0 \pm 0.0$ | $0.0 \pm 0.0$ |
| PETS-CEM | $66.9 \pm 6.3$ | $7.2 \pm 1.9$ |
| PETS-LaP$^3$ | $\mathbf{83.1 \pm 2.3}$ | $\mathbf{19.4 \pm 1.8}$ |

Table 6: Results for MiniWorld tasks for different methods using a learned PETS transition model. The oracle model is only used for final evaluation on each environment seed, and the resulting trajectory becomes future training data to the PETS model. We report the total fraction of environment seeds solved by each method, out of 256 total, averaged across 5 trials. LaP$^3$ substantially outperforms the original PETS implementation. This table corresponds to Figure 5.

| | MazeS3 | FourRooms | SelectObj |
|---|---|---|---|
| LaP$^3$ | $\mathbf{57.0 \pm 3.1}$ | $\mathbf{89.1 \pm 2.0}$ | $\mathbf{45.3 \pm 3.1}$ |
| LaP$^3$-mean | $43.4 \pm 3.1$ | $83.6 \pm 2.3$ | $16.0 \pm 2.3$ |
| LaP$^3$-nolatent | $26.6 \pm 2.8$ | $54.3 \pm 3.1$ | $2.3 \pm 0.9$ |
| LaP$^3$-notree | $24.6 \pm 2.7$ | $0.0 \pm 0.0$ | $2.7 \pm 1.0$ |
| LaP$^3$-noUCB | $28.1 \pm 2.8$ | $76.6 \pm 2.6$ | $1.6 \pm 0.8$ |

Table 7: Results for MiniWorld tasks using different region selection methods. 256 trials per method. This corresponds to Figure 6.

|         | adpcm | aes   | blowfish | dhrystone | gsm  | matmul | mpeg2 | qsort | sha    |
|---------|-------|-------|----------|-----------|------|--------|-------|-------|--------|
| -O0     | 41260 | 12633 | 199345   | 9258      | 8130 | 42085  | 10489 | 58400 | 269653 |
| -O3     | 16844 | 9937  | 188237   | 5936      | 7137 | 33244  | 8266  | 52256 | 226235 |
| PPO_50  | 11175 | 10263 | **175649** | 5753    | 6286 | **9644** | 8281 | 52137 | 209142 |
| OpenTuner | 10501 | 9795 | 180834   | 7196      | 6181 | 33244  | 8291  | 52137 | 209155 |
| CMA-ES  | **10451** | 10093 | 180198 | 5996    | 6294 | **9644** | 8280 | 50869 | 209142 |
| LaP$^3$ | **10451** | 9753 | 180179  | 5702      | **6178** | **9644** | 8282 | **47745** | 209142 |
| PPO_4000 | 10415 | 9759 | 175779  | **5515**  | 6286 | **9644** | **8260** | 47785 | **205302** |

Table 8: Results for compiler phase ordering, in execution cycles for program after a series of transformations, following setup of [15]. 50 oracle accesses per method. This table corresponds to Figure 7.

|         | QED | DRD2 | HIV | SARS |
|---------|-----|------|-----|------|
| LaMCTS  | $0.914 \pm 0.002$ | $0.323 \pm 0.016$ | $0.406 \pm 0.019$ | $0.452 \pm 0.010$ |
| RS      | $0.897 \pm 0.001$ | $0.081 \pm 0.006$ | $0.116 \pm 0.002$ | $0.279 \pm 0.006$ |
| CEM     | $0.906 \pm 0.003$ | $0.250 \pm 0.016$ | $0.455 \pm 0.021$ | $0.423 \pm 0.005$ |
| CMA-ES  | $0.888 \pm 0.004$ | $0.216 \pm 0.018$ | $0.425 \pm 0.020$ | $0.414 \pm 0.007$ |
| LaP$^3$ | $\mathbf{0.916 \pm 0.001}$ | $\mathbf{0.648 \pm 0.026}$ | $\mathbf{0.588 \pm 0.020}$ | $\mathbf{0.570 \pm 0.018}$ |

Table 9: Mean and standard deviation across 128 random seeds for LaP$^3$ and baselines on QED, DRD2, SARS, and HIV molecular design tasks; results reported for 1000, 4000, 4000, and 4000 oracle queries respectively. LaP$^3$ significantly outperforms all three baselines on all four properties. This table corresponds to Figure 8.

# F Detailed Analyses and Ablations

## F.1 $L_k$ and $c_k$ Estimation Details

To loosely approximate the Lipschitz constant in our analysis from Sec. 5.3, we simply check all pairwise Lipschitz constants between existing samples (candidate trajectories) in the tree node (region). Similarly, to loosely approximate $c_k$, we take the highest-scoring sample in the region as the "optimum" and estimate $c_k$ for $z_k = 0.5$ following our definition using the remaining samples in the region.

## F.2 $z_k$ Estimation

We estimate $z_k$ over time in our MiniWorld tasks, fixing several different values of $c_k$ in intervals of 1 reward (Figure 9). $z_k$ is estimated using 50 samples (in between each dynamic re-partitioning of the space) at each timestep in intervals of 50, with 32 trial runs. In all cases $z_k$ initially drops very quickly, and then somewhat plateaus after finding the initial local optimum (whether global or not), especially in SelectObj. However, in most cases it still continues to decrease over time.

While this is consistent with our qualitative analysis in Sec. B.5 about how $z_k$ changes with recursive splitting, in some cases, $z_k$ seems to stop decreasing over time. Upon inspection, we find that those regions whose $z_k$ remain high correspond to low-performing regions which do not receive many samples according to UCB exploration. Therefore, such regions won't improve over time and the $z_k$ remains high.

## F.3 Latent Space Visualization

We show a t-SNE visualization (Figure 10) of the latent space of trajectories at the end of a sample MazeS3 run of LaP$^3$. The first sampled trajectories are colored red, with a gradient toward blue for the later-sampled trajectories. The later trajectories are clearly separated in the latent space.

## F.4 Parameter Space Methods

It is of course possible to optimize the parameters of a policy which outputs an action given the current state, as in the original LaMCTS formulation, or in PPO. Nevertheless, we tune and run a parameter-space version of LaMCTS in the MiniWorld tasks, which is essentially the original

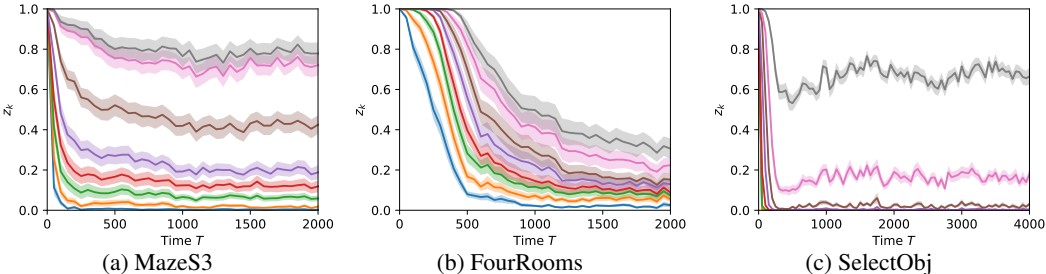

| (a) MazeS3 | (b) FourRooms | (c) SelectObj |

Figure 9: Mean and standard deviation (32 trials), of estimated $z_k$ for different values of $c_k$ on MiniWorld tasks.

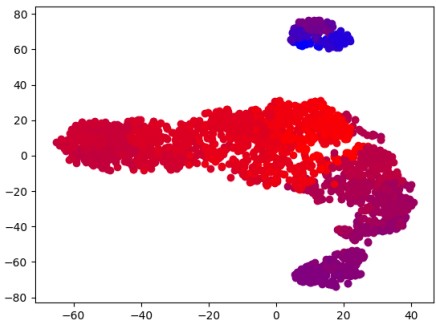

Figure 10: Latent space t-SNE visualization for a sample MazeS3 run of $\texttt{LaP}^3$. Earlier-sampled trajectories (red) are clearly separated from the latest-sampled trajectories (blue).

LaMCTS adapted to path planning, only with TuRBO replaced with CMA-ES as in $\texttt{LaP}^3$ due to speed considerations. Specifically, following LaMCTS, we learn the parameters of a linear policy for outputting actions given states.

|  | **MazeS3** | **FourRooms** | **SelectObj** |
|---|---|---|---|
| LaMCTS-parameter | $10.9 \pm 1.9$ | $9.4 \pm 1.8$ | $100.0 \pm 0.0$ |
| PPO | $0.0 \pm 0.0$ | $0.0 \pm 0.0$ | $31.3 \pm 8.2$ |
| $\texttt{LaP}^3$ | $57.0 \pm 3.1$ | $89.1 \pm 2.0$ | $45.3 \pm 3.1$ |

Table 10: Comparison of $\texttt{LaP}^3$ to an adaptation of the original LaMCTS, operating in *parameter* space, on MiniWorld tasks. SelectObj is uniquely advantageous to parameter-space methods; for this reason, PPO also performs better than our other baselines (but worse than $\texttt{LaP}^3$) on that environment only. 256 trials per method.

Working in the parameter space can be clearly advantageous when states are relatively simple and low-dimensional, as in the Mujoco environments evaluated in LaMCTS and POPLIN [47], or when the policy barely needs to depend on the state at all, as in our SelectObj task (Table 10). We designed the SelectObj task as a challenge for path planning algorithms operating in the action space, which struggle to escape the local optimum, but in truth this environment can be trivially solved by simply moving in the same correct direction at every step (toward the far goal).

On the other hand, more complex policies may be more challenging to learn when the state representation is higher-dimensional, which may be the case in practical tasks. This is the case in our MazeS3 and FourRooms environments, where the state is represented as a top-down image rather than a vector of position and velocity information. Unlike SelectObj, these tasks require navigation around obstacles rather than just moving in a straight line. Despite featurizing with the same randomly initialized CNN as $\texttt{LaP}^3$, LaMCTS-parameter performs very poorly on MazeS3 and FourRooms in comparison. Additionally, methods like LaMCTS-parameter which use a parameter space must critically depend on the specific parametric form of the policy to be learned (e.g., whether it is a linear policy, a nonlinear policy parameterized by neural networks, etc); therefore, it is not obvious

how to take advantage of a latent space which encodes a *sequence* of states and/or actions, which is critical in environments such as our molecular design tasks.

## F.5 Hyperparameter Sensitivity Analysis

Since the $C_p$ parameter is the only additional parameter we tune in $\texttt{LaP}^3$, we analyze the sensitivity of $\texttt{LaP}^3$'s performance with respect to $C_p$ on the MiniWorld tasks. Our main results use $C_p = 2$ except for SelectObj where we used $C_p = 4$; here we run $C_p = 1, 2, 4$ for all three tasks and show the results in Table 11. $\texttt{LaP}^3$ even with poorly tuned $C_p$ values still substantially outperforms CEM on these tasks with difficult-to-escape local optima.

| | MazeS3 | FourRooms | SelectObj |
|---|---|---|---|
| CEM | $25.0 \pm 2.7$ | $69.9 \pm 2.9$ | $0.4 \pm 0.4$ |
| $\texttt{LaP}^3$-$C_p = 1$ | $52.7 \pm 3.1$ | $89.5 \pm 1.9$ | $6.6 \pm 1.6$ |
| $\texttt{LaP}^3$-$C_p = 2$ | $57.0 \pm 3.1$ | $89.1 \pm 2.0$ | $23.1 \pm 2.6$ |
| $\texttt{LaP}^3$-$C_p = 4$ | $53.5 \pm 3.1$ | $87.1 \pm 2.1$ | $45.3 \pm 3.1$ |

Table 11: Results for MiniWorld tasks using different $C_p$ values for $\texttt{LaP}^3$. $C_p = 2$ corresponds to our main paper results, except for SelectObj where we used $C_p = 4$. $\texttt{LaP}^3$ is relatively insensitive to changes in $C_p$ on MazeS3 and FourRooms and only more sensitive on the more difficult SelectObj task. However, even poorly tuned versions of $\texttt{LaP}^3$ outperform CEM, reproduced for baseline comparison. 256 trials per method.

## F.6 Max vs. Mean UCB Metric For MCTS

Our theory suggests that the UCB metric for MCTS should be based on the max function value rather than the mean for the deterministic functions that we consider in this work. Figure 6 already shows this for MiniWorld; here we show the max vs. mean analysis for all tasks in Tables 12, 13, 14, and 15.

| | MazeS3 | FourRooms | SelectObj |
|---|---|---|---|
| $\texttt{LaP}^3$-mean | $43.4 \pm 3.1$ | $83.6 \pm 2.3$ | $16.0 \pm 2.3$ |
| $\texttt{LaP}^3$ | $57.0 \pm 3.1$ | $89.1 \pm 2.0$ | $45.3 \pm 3.1$ |

Table 12: Results for $\texttt{LaP}^3$ (using max function value for UCB) in MiniWorld compared to $\texttt{LaP}^3$ using the mean function value metric for UCB. $\texttt{LaP}^3$ is substantially better. 256 trials per method.

| | DK-6 | DK-8 | KC-S3R3 | KC-S3R4 | MR-N4S5 | MR-N6 |
|---|---|---|---|---|---|---|
| $\texttt{LaP}^3$-mean | $0.98 \pm 0.02$ | $0.25 \pm 0.13$ | $-2.36 \pm 0.09$ | $-4.36 \pm 0.12$ | $-11.78 \pm 0.77$ | $-114.63 \pm 4.53$ |
| $\texttt{LaP}^3$ | $0.95 \pm 0.03$ | $0.46 \pm 0.13$ | $-2.27 \pm 0.09$ | $-4.37 \pm 0.13$ | $-11.68 \pm 0.75$ | $-113.53 \pm 4.49$ |

Table 13: Results for $\texttt{LaP}^3$ (using max function value for UCB) in MiniGrid compared to $\texttt{LaP}^3$ using the mean function value metric for UCB. $\texttt{LaP}^3$ performs similarly or better. 256 trials per method.

| | adpcm | aes | blowfish | dhrystone | gsm | matmul | mpeg2 | qsort | sha |
|---|---|---|---|---|---|---|---|---|---|
| $\texttt{LaP}^3$-mean | 10501 | 10407 | 176429 | 5740 | 6305 | 8841 | 8281 | 47745 | 209142 |
| $\texttt{LaP}^3$ | 10451 | 9753 | 180179 | 5702 | 6178 | 9644 | 8282 | 47745 | 209142 |

Table 14: Results for $\texttt{LaP}^3$ (using max function value for UCB) in compiler phase ordering compared to $\texttt{LaP}^3$ using the mean function value metric for UCB. The two versions perform similarly. 256 trials per method.

## F.7 Latent Space Ablations

We conduct additional analysis on the use of a latent partition space $\Phi_s$ in the MiniWorld, MiniGrid, and compiler phase ordering tasks in Tables 16 (reproduced from Table 7), 17, and 18 respectively.

|  | QED | DRD2 | HIV | SARS |
|---|---|---|---|---|
| LaP$^3$-mean | $0.914 \pm 0.002$ | $0.323 \pm 0.016$ | $0.406 \pm 0.019$ | $0.452 \pm 0.010$ |
| LaP$^3$ | $0.916 \pm 0.001$ | $0.648 \pm 0.026$ | $0.588 \pm 0.020$ | $0.570 \pm 0.018$ |

Table 15: Results for LaP$^3$ (using max function value for UCB) in molecular design tasks compared to LaP$^3$ using the mean function value metric for UCB. LaP$^3$ performs similarly on the easiest QED task and substantially better on the others. 256 trials per method.

|  | MazeS3 | FourRooms | SelectObj |
|---|---|---|---|
| LaP$^3$-nolatent | $26.6 \pm 2.8$ | $54.3 \pm 3.1$ | $2.3 \pm 0.9$ |
| LaP$^3$ | $57.0 \pm 3.1$ | $89.1 \pm 2.0$ | $45.3 \pm 3.1$ |

Table 16: Results for LaP$^3$ in MiniWorld compared to LaP$^3$ without the use of a partition latent space $\Phi_s$. LaP$^3$ is substantially better. 256 trials per method.

|  | DK-6 | DK-8 | KC-S3R3 | KC-S3R4 | MR-N4S5 | MR-N6 |
|---|---|---|---|---|---|---|
| LaP$^3$-nolatent | $0.98\pm0.02$ | $0.25\pm0.13$ | $-2.36\pm0.09$ | $-4.36\pm0.12$ | $-11.78\pm0.77$ | $-114.63\pm4.53$ |
| LaP$^3$ | $0.95\pm0.03$ | $0.46\pm0.13$ | $-2.27\pm0.09$ | $-4.37\pm0.13$ | $-11.68\pm0.75$ | $-113.53\pm4.49$ |

Table 17: Results for LaP$^3$ in MiniGrid compared to LaP$^3$ without the use of a partition latent space $\Phi_s$. LaP$^3$ is better in most cases. 256 trials per method.

|  | adpcm | aes | blowfish | dhrystone | gsm | matmul | mpeg2 | qsort | sha |
|---|---|---|---|---|---|---|---|---|---|
| LaP$^3$-nolatent | 10451 | 10263 | 176429 | 6617 | 6169 | 8841 | 8280 | 52137 | 476269 |
| LaP$^3$ | 10451 | 9753 | 180179 | 5702 | 6178 | 9644 | 8282 | 47745 | 209142 |

Table 18: Results for LaP$^3$ in compiler phase ordering compared to LaP$^3$ without the use of a partition latent space $\Phi_s$. The two versions are comparable in most cases.

LaP$^3$ performs similarly or better compared to the version without a latent space; the difference is especially large in MiniWorld.

Additionally, it is possible to use a separate sampling latent space $\Phi_h$, as illustrated here in MiniGrid (Table 19), although we do not do so in our main results to keep consistency between latent spaces across tasks. The version with a latent space (a reversible flow model here) performs slightly better.

|  | DK-6 | DK-8 | KC-S3R3 | KC-S3R4 | MR-N4S5 | MR-N6 |
|---|---|---|---|---|---|---|
| LaP$^3$-latent$\Phi_h$ | $0.97\pm0.02$ | $0.48\pm0.11$ | $-2.19\pm0.15$ | $-4.22\pm0.13$ | $-10.68\pm0.68$ | $-112.72\pm4.46$ |
| LaP$^3$ | $0.95\pm0.03$ | $0.46\pm0.13$ | $-2.27\pm0.09$ | $-4.37\pm0.13$ | $-11.68\pm0.75$ | $-113.53\pm4.49$ |

Table 19: Results for LaP$^3$ in MiniGrid compared to LaP$^3$ with the use of a sampling latent space. While the differences are small on most cases, LaP$^3$ with a latent $\Phi_h$ is better on all tasks. 256 trials per method.

We additionally ablate on the latent space (used for both partitioning and sampling) in the easiest of our molecular design tasks, the QED property. Specifically, we build the molecular SMILES string autoregressively, using a discrete action space with 10 choices: the 9 most common characters in molecular SMILES strings, in addition to an end token. (We limit the space of possible characters in order to increase the chances of generating well-formed SMILES strings.) We optimize in a continuous space of action probabilities as in MiniGrid, and allow a maximum length of 50 characters.

The poor results demonstrate the absolute *necessity* of a latent space in the molecular design task (Table 20). While typical molecular SMILES strings for this task are 30 to 50 characters long, both LaP$^3$ and baselines struggle to generate well-formed strings even of length 3 to 5 without the pre-trained latent space. Accordingly, the performance is drastically lower for all methods.

|         | QED |
|---------|-----|
| RS-no$\Phi$ | $0.417 \pm 0.002$ |
| CEM-no$\Phi$ | $0.411 \pm 0.002$ |
| CMA-ES-no$\Phi$ | $0.403 \pm 0.003$ |
| LaP$^3$-no$\Phi$ | $0.416 \pm 0.003$ |
| RS | $0.897 \pm 0.001$ |
| CEM | $0.906 \pm 0.003$ |
| CMA-ES | $0.888 \pm 0.004$ |
| LaP$^3$ | $0.916 \pm 0.001$ |

Table 20: Mean and standard deviation across 128 random seeds for LaP$^3$ and baselines on QED, with and without the pre-trained latent space.

## F.8  Other Inner Solvers

In this work we have used CMA-ES as the inner solver due to its speed and acceptable performance. The original LaMCTS work used the TuRBO solver [10], which is prohibitively slow for many of our experiments. Nevertheless, we have run experiments on the MiniWorld tasks using a smaller number of trials to check performance using an alternate inner solver, both on LaP$^3$ and also on the LaMCTS baseline (Table 21. In most cases TuRBO performs equal or worse; we hypothesize this is because our tasks use a smaller query budget per trial compared to the original LaMCTS work, causing TuRBO to use too large a fraction of its total budget in each inner loop.

|         | MazeS3 | FourRooms | SelectObj |
|---------|--------|-----------|-----------|
| LaMCTS-TuRBO | $21.9 \pm 7.3$ | $0.0 \pm 0.0$ | $0.0 \pm 0.0$ |
| LaMCTS | $23.4 \pm 2.6$ | $20.3 \pm 2.5$ | $0.8 \pm 0.6$ |
| LaP$^3$-TuRBO | $52.2 \pm 10.4$ | $63.6 \pm 14.5$ | $47.1 \pm 12.1$ |
| LaP$^3$ | $57.0 \pm 3.1$ | $89.1 \pm 2.0$ | $45.3 \pm 3.1$ |

Table 21: Results for LaP$^3$ and LaMCTS with CMA-ES and TuRBO inner solvers in MiniWorld, with fewer trials for TuRBO due to computational expense. TuRBO generally performs equal or worse on this task.

## G  Baseline Details and Hyperparameter Tuning

LaP$^3$. For our method, we try $C_p$ in $\{0.5, 1, 2, 4\}$. If the search space is not explicitly bounded, we sample the first $N_{init}$ points used to initialize the partition tree using the same $\sigma$ as CEM. Note $N_{init}$ is not tuned; we use 5 for compiler phase optimization where our query budget is only 50, and 50 elsewhere. No other hyperparameters are tuned.

**LaMCTS.** Detailed in Algorithm 1, where we summarize the changes made in LaP$^3$ compared to LaMCTS. It is tuned similarly to our own method LaP$^3$.

**RS.** The simplest baseline, in which one simply samples random trajectories and in the end returns the best-performing among them. We do not tune this baseline.

**CEM.** An evolutionary method which tracks a population of $N$ samples. At each step, it selects the best $N_e$ samples from its population to initialize the mean $\mu$ for the next generation of $N$ samples, drawn from a Gaussian distribution with standard deviation $\sigma$. However, while too-small $\sigma$ may prevent CEM from escaping local optima, too-large $\sigma$ may yield results little better than random shooting. We find that the choice of $\sigma$ is critical to CEM's performance in our test environments. Therefore, we systematically tune $\sigma$ when running CEM in all environments (checking $\{1, 2, 4, 8\}$). While other parameters such as $N$ and $N_e$ are also tunable, we find that these make a smaller difference, so we did not tune them extensively.

**CMA-ES.** A more complex evolutionary method which can be viewed as a variant of CEM. After providing an initial $\mu$ and $\sigma$ for the first generation, CMA-ES determines its own $\sigma$ automatically afterward, while also fitting additional parameters. Even so, we find that its performance is highly sensitive to the initial $\sigma$, and we tune this parameter in the same way that we do for CEM.

**VOOT.** An MCTS method which builds a tree on actions. We tune the exploration parameter in the VOO submodule, trying values in $\{0.1, 0.3, 0.5\}$.

**RandDOOT.** An MCTS method which builds a tree on actions similar to VOOT, but which splits using axis-aligned boundaries rather than splitting into Voronoi regions; used as a baseline in their original paper [22]. We did not tune hyperparameters.

**iLQR.** A gradient-based optimization method for continuously optimizing the planned trajectory, which we run to convergence. It cannot easily escape local optima. As the performance was relatively insensitive to hyperparameters, we did not systematically tune.

**PPO.** A standard reinforcement learning algorithm which operates in the parameter space, unlike our other baselines. Since PPO is relatively robust to hyperparameters [39], we didn't systematically tune.

## H   Additional Environment Details

### H.1   MiniWorld

We modified the original MiniWorld environments to have continuous action spaces and to have more consistent difficulty across random seeds, as follows.

**MazeS3**. A 3x3 maze of rooms which are each 3 units by 3 units, with walls between rooms being 0.25 units wide. The maze is constructed by recursive backtracking from the top left room. The agent begins in the top left room and the goal is placed in the last room generated in the maze construction. The step size is 0.3 units and the environment length is 216 steps. The final sparse reward is the Euclidean distance between the agent and the goal if the goal is not reached, otherwise a fixed reward of 1 penalized by a fraction of the number of steps taken, down to a minimum of 0.8.

**FourRooms**. A 14x14 unit space with a 6x6 room in each corner. Adjacent rooms are connected by a width-2 corridor along the outer edge of the space, i.e., there is a cross-shaped obstacle in the center. The agent starts in a random location and the goal is in the diametrically opposite location. The step size is 0.2 units and the environment length is 250 steps. The final sparse reward is the Euclidean distance between the agent and the goal if the goal is not reached, otherwise a fixed reward of 1 penalized by a fraction of the number of steps taken, down to a minimum of 0.8.

**SelectObj**. A 12x12 unit open space. The agent starts in the center. The near goal is 4 to 4.5 units away and the far goal is 5 to 5.5 units away. The two goals are 3 to 4 units away from each other. The step size is 0.05 units and the environment length is 200 steps. Unlike MazeS3 and FourRooms, SelectObj does not terminate upon reaching a goal. The final sparse reward is the Euclidean distance between the final agent position and the closest goal to the final position, plus a fixed reward of 1 for being within 1 unit of the original far goal.

### H.2   Compiler Phase Ordering

The action space consists of 46 different program transformations, and a trajectory consists of 45 transformations (quite short, considering many transformations have no effect unless applied in a specific order). The reward is the difference between the original and final number of execution cycles. Since the environment is deterministic, we only run 1 trial for each method. Thus far we have followed the setup in [15]; however, unlike [15], we allow a budget of only 50 trajectory queries.