# OpenReview forum: "Learning Space Partitions for Path Planning"
_NeurIPS.cc/2021/Conference — NeurIPS 2021 Poster_

### Official Review · Reviewer_DXgW · 2021-07-05

**Rating:** 7
**Confidence:** 3

**Summary:**

The paper proposes a new path planning algorithm that extends an existing algorithm in two ways: a) using a latent space for dimensionality reduction b) using the maximum observed reward instead of the mean, which allows for faster convergence in deterministic environments. A theoretical analysis is provided to justify the latter modification.

**Ethical Concerns:**

No ethical concerns.

**Limitations And Societal Impact:**

No concerns here.

**Main Review:**

The paper is good at establishing the ideas of space-partitioning, as well as the difference between adaptive and non-adaptive partitioning methods.

I must admit that I have difficulty following the theoretical analysis. Specifically the concept of $(z_k, c_k)$-dilution is not very clear to me. Perhaps an intuitive explanation would be helpful here.

The experiments section is quite extensive and covers a variety of settings. The proposed method performs favorably in each scenario.

That being said, I'm concerned with the baselines. I would have expected a comparison to La-MCTS, since it is the closest related work. The closest thing to a comparison to La-MCTS are the ablations on MiniWorld, where the mean is used instead of the max. That at least covers one of the two modifications proposed, but the other is missing (using a latent space for partitioning).

Another issue is that La-MCTS was originally tested on a different experiment suite, which has no overlap with the experiments in this paper. Since this paper is proposing modifications on top of La-MCTS, I believe that a comparison on at least some of the experiments of La-MCTS' original experiment section is necessary.

Overall I think the paper is promising and I strongly encourage the authors to continue this line of work. Unfortunately, I believe that the comparison to La-MCTS is currently not very clear, which is the main reason why I see this paper as "marginally above the threshold". I would definitely revise my opinion if the authors could address the following points:

* Provide a comparison to La-MCTS. If possible across the entire experiment suite in this paper.
* Compare to La-MCTS on some of the experiments that are in the experiments section of La-MCTS.

I think the paper would be a clear accept if these could be fulfilled and the comparison is generally in PlaLaM's favor.

**Other comments**

* The paper seems to focus on deterministic reward functions. I think it's worth discussing or at least mentioning this as a limitation, since taking the maximum might under-perform compared to the mean under stochastic rewards.

* The paper never explains what "periodic snapshots" are. I'm not familiar with this term. I think it should be introduced somewhere in the text.

**Edit following author response**

Given the favorable comparison to La-MCTS, I've increased my rating from 6 to 7. Thank you to the authors for providing such extensive additional results in a short time frame!

**Time Spent Reviewing:**

2

---

> ### Author Response · Authors · 2021-08-10
> **Reply to Reviewer DXgW**
>
> Thank you for your helpful comments! Regarding your concerns:
>
> - **Intuitive explanation of $(z_k, c_k)$-dilution**
>
> The intuition for dilution for a given region, as depicted in Fig 2a, is that all but $z_k$ fraction of the region has function value close to the maximum, with "close" defined based on
> $c_k$ (smaller $c_k$ = stricter definition of "close"). A less diluted region (i.e., $z_k$ and $c_k$ are both small) means that function values in the most part of the region are concentrated near the maximum and thus easy to optimize. We will add the intuition to the text.
>
>
> - **Comparison to LaMCTS**
>
> Thanks for the suggestion! We have added this comparison on all of our tasks, and PlaLaM generally outperforms LaMCTS; please see the general comments.
>
>
> - **Comparison to LaMCTS on some of LaMCTS paper’s original tasks**
>
> We have run experiments on the Ackley and Levy function optimization tasks tested in the LaMCTS paper, doing both 20- and 100-dimensional versions of each. PlaLaM performs equal to or better than LaMCTS on these tasks (*lower* is better).
>
> |        | Ackley-20D    | Levy-20D      | Ackley-100D   | Levy-100D      |
> |--------|---------------|---------------|---------------|----------------|
> | LaMCTS | 0.48 +/- 0.03 | 0.51 +/- 0.09 | 0.65 +/- 0.25 | 14.24 +/- 4.87 |
> | PlaLaM | 0.49 +/- 0.04 | 0.34 +/- 0.07 | 0.46 +/- 0.15 | 11.95 +/- 3.56 |
>
>
> Specifically, for the 20-dimensional tasks, both solvers are using TuRBO as the inner solver following the setup in the original LaMCTS paper, and we use 1000 sample budget like the original paper. For the 100-dimensional tasks, we switch to CMA-ES as the inner solver and only run 4000 sample budget instead of 10000, due to computational burden. All runs are 8 trial seeds per method.
>
> The original LaMCTS paper also runs Mujoco tasks, but they optimize a *policy* rather than a *trajectory*. We leave the parameter space Mujoco experiments to future work.
>
> - **Focus on deterministic reward functions**
>
> We do focus on deterministic reward functions; thanks for pointing this out. This is indeed a limitation of PlaLaM. We will make our focus on deterministic reward functions more explicit in the revision.
>
>
> - **Definition of “periodic snapshots”**
>
> When we say periodic snapshots of the state in the 2D navigation and compiler tasks, we mean that we collect all of the observed states (what we call "snapshots") over the course of the full trajectory, and then pick every n^th state and discard the rest (e.g., n=20 in 2D navigation tasks) to reduce overall dimensionality. The remaining states are concatenated together as the latent space for partitioning. We'll explain this more clearly in the text.

---

> > ### Comment · Reviewer_DXgW · 2021-08-12
> > **Thank you for the additional results!**
> >
> > I've looked at the additional results the authors have posted throughout the thread and have increased my score accordingly (see "Edit following author response"). I believe the paper should be accepted.

---

> > > ### Author Response · Authors · 2021-08-12
> > > **Thank you!**
> > >
> > > Thanks again for your insightful review!

---

### Official Review · Reviewer_vamd · 2021-07-12

**Rating:** 7
**Confidence:** 3

**Summary:**

The paper presents a black-box optimization method that efficiently searches the parameter by partitioning the parameter space using a latent space. The proposed method is built upon LaMCTS proposed by Wang et al. [2020]. This paper provides theoretical analysis of LaMCTS and proposed the modifications. The proposed algorithm, PlaLaM, was tested in synthetic tasks and real-world tasks, including compiler optimization and molecular design tasks. The proposed algorithm outperformed baseline algorithms in those experiments.

**Limitations And Societal Impact:**

Although the proposed method exhibits promising performance in this study, I do not think that the proposed method always outperforms CMA-ES because PlaLaM itself uses CMA-ES in its inner process. I recommend to clarify when to employ PlaLaM instead of CMA-ES and vice versa, which would be useful for practitioners.

**Main Review:**

Strong points:
-	Theoretical analysis of LaMCTS provides a better understanding of LaMCTS.
-	The proposed modification is grounded on the theoretical analysis.

Weak points:
-	Although the proposed algorithm PlaLaM outperformed well-known baseline methods such as CEM and CMA-ES, comparison with LaMCTS is not provided.

I think that the paper is overall well-written, and the theoretical analysis of LaMCTS is beneficial to the community. My main and only concern is that the proposed algorithm, PlaLaM is not compared with LaMCTS in the experiments, if I understand correctly. As the main contribution of the paper is analysis of LaMCTS and its modifications, the comparison between LaMCTS and PlaLaM is crucial to support the claims in the paper.

I recommend authors to add ablation study to investigate the effect of the proposed modifications, e.g., evaluating PlaLaM with mean values like LaMCTS, PlaLaM without a latent model, and PlaLam with TuRBO instead of CMA-ES. If the ablation study is added, I'm happy to increase the score.

=== comments after the author response ===

The additional results provided by authors clearly demonstrate that how PlaLaM outperforms LaMCTS. I updated the score.



**Time Spent Reviewing:**

3

---

> ### Author Response · Authors · 2021-08-10
> **Reply to Reviewer vamd**
>
> Thank you for your helpful comments! Regarding your concerns:
>
> &nbsp;
>
> - **Comparison to LaMCTS**
>
> Thanks for the suggestion! We have added this comparison on all of our tasks, and PlaLaM generally outperforms LaMCTS; please see the general comments.
>
> &nbsp;
>
> - **Ablation Studies**
>
> Thanks, we agree that additional ablation studies would be useful. We have run the three ablations you suggested ((1) mean values instead of max, (2) without latent model, (3) TuRBO inner solver instead of CMA-ES). PlaLaM generally outperforms the ablated versions, justifying our design choices.
>
> **--Ablation (1): PlaLaM with mean values instead of max--**
>
> **MiniWorld** (previously reported in Sec. 5.3):
>
> |             | MazeS3       | FourRooms    | SelectObj    |
> |-------------|--------------|--------------|--------------|
> | PlaLaM-mean | 43.4 +/- 3.1 | 83.6 +/- 2.3 | 16.0 +/- 2.3 |
> | PlaLaM      | 57.0 +/- 3.1 | 89.1 +/- 2.0 | 45.3 +/- 3.1 |
>
>
> **MiniGrid:**
>
> |             | DK-6          | DK-8          | KC-S3R3        | KC-S3R4        | MR-N4S5         | MR-N6            |
> |-------------|---------------|---------------|----------------|----------------|-----------------|------------------|
> | PlaLaM-mean | 0.98 +/- 0.02 | 0.25 +/- 0.13 | -2.36 +/- 0.09 | -4.36 +/- 0.12 | -11.78 +/- 0.77 | -114.63 +/- 4.53 |
> | PlaLaM      | 0.95 +/- 0.03 | 0.46 +/- 0.13 | -2.27 +/- 0.09 | -4.37 +/- 0.13 | -11.68 +/- 0.75 | -113.53 +/- 4.49 |
>
>
> **Compiler Optimization:**
>
> |             | adpcm | aes  | blowfish | dhrystone | gsm  | matmul | mpeg2 | qsort | sha  |
> |-------------|-------|------|----------|-----------|------|--------|-------|-------|------|
> | PlaLaM-mean | 1.00  | 0.94 | 1.02     | 1.25      | 0.98 | 3.76   | 1.00  | 1.09  | 1.00 |
> | PlaLaM      | 1.00  | 1.00 | 1.00     | 1.26      | 1.00 | 3.45   | 1.00  | 1.09  | 1.00 |
>
>
> **Molecular Design:**
>
> |              | QED             | DRD2            | HIV             | SARS            |
> |--------------|-----------------|-----------------|-----------------|-----------------|
> | PlaLaM-mean* | 0.914 +/- 0.002 | 0.323 +/- 0.016 | 0.406 +/- 0.019 | 0.452 +/- 0.010 |
> | PlaLaM       | 0.916 +/- 0.001 | 0.648 +/- 0.026 | 0.588 +/- 0.020 | 0.570 +/- 0.018 |
>
> *this is the same as the comparison to LaMCTS in the general comments, because we use the pretrained latent space in LaMCTS for fair comparison
>
> **--Ablation (2): PlaLaM without latent model--**
>
> **MiniWorld:** (previously reported in Appendix E.5)
>
> |                  | MazeS3       | FourRooms    | SelectObj    |
> |------------------|--------------|--------------|--------------|
> | PlaLaM-no latent | 26.6 +/- 2.8 | 54.3 +/- 3.1 | 2.3 +/- 0.9  |
> | PlaLaM           | 57.0 +/- 3.1 | 89.1 +/- 2.0 | 45.3 +/- 3.1 |
>
>
>
> **MiniGrid:**
>
>
> |             | DK-6          | DK-8          | KC-S3R3        | KC-S3R4        | MR-N4S5         | MR-N6            |
> |-------------|---------------|---------------|----------------|----------------|-----------------|------------------|
> | PlaLaM-mean | 0.96 +/- 0.03 | 0.32 +/- 0.14 | -2.54 +/- 0.09 | -4.40 +/- 0.12 | -11.10 +/- 0.70 | -115.78 +/- 4.57 |
> | PlaLaM      | 0.95 +/- 0.03 | 0.46 +/- 0.13 | -2.27 +/- 0.09 | -4.37 +/- 0.13 | -11.68 +/- 0.75 | -113.53 +/- 4.49 |
>
>
> **Compiler Optimization:**
>
>
> |             | adpcm | aes  | blowfish | dhrystone | gsm  | matmul | mpeg2 | qsort | sha  |
> |-------------|-------|------|----------|-----------|------|--------|-------|-------|------|
> | PlaLaM-mean | 1.00  | 0.95 | 1.02     | 1.08      | 1.00 | 3.76   | 1.00  | 1.00  | 0.44 |
> | PlaLaM      | 1.00  | 1.00 | 1.00     | 1.26      | 1.00 | 3.45   | 1.00  | 1.09  | 1.00 |
>
> **Molecular Design:**
>
> (Omitted since we use a pretrained latent space, without which all methods struggle to generate valid molecules of reasonable length.)
>
> **--Ablation (3): PlaLaM with TuRBO instead of CMA-ES--**
>
> **MiniWorld:**
>
> |              | MazeS3        | FourRooms     | SelectObj     |
> |--------------|---------------|---------------|---------------|
> | PlaLaM-TuRBO | 52.2 +/- 10.4 | 63.6 +/- 14.5 | 47.1 +/- 12.1 |
> | PlaLaM       | 57.0 +/- 3.1  | 89.1 +/- 2.0  | 45.3 +/- 3.1  |
>
>
> We only run the TuRBO ablation on MiniWorld, and with fewer trial seeds, due to the computational burden of TuRBO. Note we use CMA-ES primarily due to computational burden, not necessarily to improve performance.
>
> &nbsp;
>
> - **When to employ PlaLaM vs CMA-ES in practice**
>
> Note that PlaLaM is a meta-algorithm which can generally improve over solvers used in the inner process (e.g., CMA-ES) by facilitating improved exploration, as shown in our experiments. Therefore, our focus is whether PlaLaM+CMA-ES can do better than CMA-ES in different scenarios, rather than directly comparing a “generic” PlaLaM with CMA-ES. Similarly, the original LaMCTS paper demonstrates significant improvements over the base TuRBO solver in their inner process.
>
> However, in a scenario where good exploration is not needed to achieve strong performance, or where the optimization landscape is smooth without suboptimal local optima, it's likely that a meta-algorithm like PlaLaM would not be required to improve over the inner CMA-ES solver. PlaLaM is likely to be most useful when the planning objective contains many local minima.

---

> > ### Comment · Reviewer_vamd · 2021-08-11
> > **Thank you for the reply**
> >
> > The experimental results provided in the response resolved my concerns in the initial review. I appreciate the authors' efforts for adding new results in a limited time.

---

> > > ### Author Response · Authors · 2021-08-11
> > > **Thank you!**
> > >
> > > Thanks again for the great suggestions for improving the paper!

---

### Official Review · Reviewer_vLZD · 2021-07-13

**Rating:** 7
**Confidence:** 3

**Summary:**

This paper introduces the algorithm PlaLaM, which extends Latent Space Monte Carlo Tree Search (LaMCTS) for tasks ranging from generating paths for mazes, to finding molecular design. The authors explore the partition of LaMCTS, and provide theoretical evidence that lower-dimensional and smoother representations lead to lower regret.

**Limitations And Societal Impact:**

Limitations: design choices for different environments should be discussed.
Societal Impact: N/A

**Main Review:**

Strengths:
The authors test their algorithm on a variety of synthetic and real world planning problems and provide better results with different baselines. The algorithm’s ability to plan in large state-space is one of the biggest strengths of this method. The authors also provide empirical evidence for their choice of design for choosing maximum as a measure of goodness, and data-driven partitions.

Weaknesses:
The choice of the transformer function for the partition and sampling space seems arbitrary. For example, it is not clear why for the Maze3D environment an identity transformer was used for the sample space while for the MiniGrid environment a reversible flow model is used.

Clarity:
The algorithm itself is very well explained, but it was hard to follow the various setups for the different experiments.

Relation to Prior Work:
PlaLaM is built upon LaMCTS, but uses the maximum value rather than the mean for the measure of goodness for each partition.

Additional Feedback:
- Visualization of the state space for the synthetic environments would be nice.
- More specifics or explanation about the results for Compiler Phase ordering can be provided since not all experiments have 61%-245% speedup.
- In figure 3, the goal location is reported incorrectly as a square

Update (after revision): Increasing score!


**Time Spent Reviewing:**

6

---

> ### Author Response · Authors · 2021-08-10
> **Reply to Reviewer vLZD**
>
> Thank you for your helpful comments! Regarding your concerns:
>
> - **Choice of transformer function for partition and sampling space**
>
> We originally chose to disentangle the partition and sampling latent spaces in order to present the most general formulation of our method, and to demonstrate that choices could in principle be made independently for each.
>
> We have tried to keep the choice of transformations as consistent as possible. For the partition latent space, it is held fixed within the same domain (2D navigation, compiler phase ordering and molecular design), and only differs between completely different domains. We emphasize that we tested on widely different domains whose observations range from image input (2D navigation) to code statistics (compiler phase ordering) and graph embeddings (molecular design), in order to show the wide applicability of our proposed method. Therefore, the choice of transformation needs to be different.
>
> For the sampling space, in the revision we will just use the identity in our experiments. The only exception is in the molecule experiments, where we take advantage of a pre-trained latent space for both partitioning and sampling.
>
> Reversible flow sampling space: while the results on MiniGrid with reversible flow sampling are  higher, the results with the identity are not much worse (Appendix E.5, reproduced here for convenience):
>
> |                                                        | DK-6          | DK-8          | KC-S3R3        | KC-S3R4        | MR-N4S5         | MR-N6            |
> |--------------------------------------------------------|---------------|---------------|----------------|----------------|-----------------|------------------|
> | PlaLaM reversible flow (submission numbers)            | 0.97 +/- 0.02 | 0.48 +/- 0.11 | -2.19 +/- 0.15 | -4.22 +/- 0.13 | -10.68 +/- 0.68 | -112.72 +/- 4.46 |
> | PlaLaM no reversible flow (new numbers moving forward) | 0.95 +/- 0.03 | 0.46 +/- 0.13 | -2.27 +/- 0.09 | -4.37 +/- 0.13 | -11.68 +/- 0.75 | -113.53 +/- 4.49 |
> | Best other baseline for each task*                     | 0.97 +/- 0.01 | 0.34 +/- 0.13 | -2.38 +/- 0.09 | -4.27 +/- 0.12 | -14.31 +/- 0.78 | -117.50 +/- 4.61 |
>
>
> *(Random shooting for first 4 tasks, CMA-ES for last 2)
>
>
> - **Visualization of the state space for the synthetic environments**
>
> Thanks for the suggestion! We created a visualization where samples in the latent partition space are mapped to 2 dimensions via t-SNE and then colored based on their position in the binary tree (since each non-leaf node in the partition tree has a high-reward child and a low-reward child). Visualization for one MazeS3 run attached at https://imgur.com/a/AgwMW4y (red = lower reward subtrees, blue = high-reward subtrees); the highest-reward subtrees are clearly separated in the t-SNE.
>
>
>
> - **Other suggestions (specifics about compiler experiments; figure caption errors)**
>
> Thank you, we will clarify/fix the descriptions.

---

> > ### Comment · Reviewer_vLZD · 2021-08-11
> > **Thank you for the revisions.**
> >
> > The authors have addressed my concerns. Overall, the paper looks good.

---

> > > ### Author Response · Authors · 2021-08-12
> > > **Thank you!**
> > >
> > > Great, thanks again for your valuable feedback!

---

### Official Review · Reviewer_rxG7 · 2021-07-16

**Rating:** 7
**Confidence:** 2

**Summary:**

This work proposes a new path planning method PlaLaM, which is an extension of the LaMCTS algorithm.
The authors provide a novel regret analysis of adaptive region partitioning schemes.

The authors demonstrate that the PlaLam method improves performance in toy 2D navigation tasks. The approach is also be applied to compiler phase ordering and molecular design achieving comparable performance to a solution based on proximal policy optimization at 4000 episodes.

The authors show comparisons to a number of baseline methods, but do not include a comparison the the LaMCTS method, which there approach is an extension of. I look forward to their explanation in the rebuttal.

Model-free RL and planning are my domains of expertise. Evolutionary algorithms are not, so please consider my review according.

**Limitations And Societal Impact:**

I am not sure why the authors do no include a comparison to LaMCTS, surely this is the most comparable method. I look forward to their explanation in the rebuttal.

**Main Review:**

Strengths:
1. The paper is well written, with clear and detailed explainations.
2. The authors provide a reference codebase which they plan to share with the community
3. The changes over the LaMCTS are clearly descrive in algoritm 1.


Weaknesses
1. The method seems an incremental improvement of the LaMCTS algorithm, is Neurips the most appropriate venue for such a work?
2. The method is comparable or outperformed by the PPO-4000 results, what advantage does this method have, sample efficiency?
3. For the 2D navigation problems, why not compare against a RL approach in addition to  ES?
4. In the all of the experiments, the authors do not include a comparison to LaMCTS, why is the comparison not possible?

Update: The authors answers my questions and addressed my main concern of a comparison with LaMCTS. Score raised to 7.


**Time Spent Reviewing:**

2

---

> ### Author Response · Authors · 2021-08-10
> **Reply to Reviewer rxG7**
>
> Thank you for your helpful comments! Regarding your concerns:
>
> **1. Improvement over the original LaMCTS algorithm**
>
> *We have made substantial contributions compared to the original LaMCTS algorithm.* As demonstrated empirically in our new comparisons to LaMCTS (please see general comments), our improvements over LaMCTS dramatically impact performance in many path planning tasks. Another piece of our contribution (as you noted) is that we provide a novel regret analysis of adaptive space partitioning, which is not present in LaMCTS. Moreover, this theory is closely linked to our algorithmic choices in practice (max vs mean observed reward, use of latent space).
>
> **2. Advantage over PPO-4000**
>
> Sample efficiency is the difference, as you suggest. PPO-4000, using 4000 sample budget, is intended as an "oracle" comparison (we got the number from [1] who ran the experiment for much longer). PlaLaM and the "fair" baselines in our setup allow only 50 samples.
>
> [1] Huang, Qijing, et al. "AutoPhase: Juggling HLS Phase Orderings in Random Forests with Deep Reinforcement Learning." arXiv preprint arXiv:2003.00671 (2020).
>
> **3. Comparison with RL approach in 2D navigation problems**
>
> We have run a standard implementation of PPO on the three MiniWorld 2D navigation tasks (only 32 trials rather than 256 due to computational burden). PPO is given the same sample budget as PlaLaM (2000 in MazeS3 and FourRooms, 4000 in SelectObj), but performs worse compared to our method PlaLaM:
>
> |        | MazeS3       | FourRooms    | SelectObj    |
> |--------|--------------|--------------|--------------|
> | PPO    | 0.0 +/- 0.0  | 0.0 +/- 0.0  | 31.3 +/- 8.2 |
> | PlaLaM | 57.0 +/- 3.1 | 89.1 +/- 2.0 | 45.3 +/- 3.1 |
>
>
> PPO’s poor performance is perhaps explained by the long time horizon (hundreds of steps) combined with a sparse reward being given only at the end of each trajectory.
>
> **4. Comparison to LaMCTS**
>
> Thanks for the suggestion! We have added this comparison on all of our tasks, and PlaLaM generally outperforms LaMCTS; please see the general comments.

---

### Author Response · Authors · 2021-08-10
**General Comment to Reviewers**

We thank all reviewers for their detailed feedback. We appreciate the comments that the paper is well written (rxG7, vamd), with a novel theoretical analysis of space partitioning which informs our design choices (vLZD, vamd, DXgW).

Reviewers (rxG7, vamd, DXgW) asked how PlaLaM compares to LaMCTS, and we have now run this experiment. Empirically, PlaLaM outperforms LaMCTS* across our experiment suite, often substantially.

**MiniWorld** (higher is better for all tasks):

|        | MazeS3       | FourRooms    | SelectObj    |
|--------|--------------|--------------|--------------|
| LaMCTS | 23.4 +/- 2.6 | 20.3 +/- 2.5 | 0.8 +/- 0.6  |
| PlaLaM | 57.0 +/- 3.1 | 89.1 +/- 2.0 | 45.3 +/- 3.1 |

**MiniGrid:**

|          | DK-6          | DK-8          | KC-S3R3        | KC-S3R4        | MR-N4S5         | MR-N6            |
|----------|---------------|---------------|----------------|----------------|-----------------|------------------|
| LaMCTS   | 0.96 +/- 0.02 | 0.09 +/- 0.17 | -2.63 +/- 0.09 | -4.43 +/- 0.13 | -14.71 +/- 0.87  | -118.70 +/- 4.68 |
| PlaLaM** | 0.95 +/- 0.03 | 0.46 +/- 0.13 | -2.27 +/- 0.09 | -4.37 +/- 0.13 | -11.68 +/- 0.75 | -113.53 +/- 4.49 |


**Compiler Optimization** (normalized speedup in cycles compared to OpenTuner):

|        | adpcm | aes  | blowfish | dhrystone | gsm  | matmul | mpeg2 | qsort | sha  |
|--------|-------|------|----------|-----------|------|--------|-------|-------|------|
| LaMCTS | 0.25  | 1.00 | 1.02     | 1.23      | 0.93 | 3.45   | 1.00  | 1.02  | 1.00 |
| PlaLaM | 1.00  | 1.00 | 1.00     | 1.26      | 1.00 | 3.45   | 1.00  | 1.09  | 1.00 |


**Molecular Design:**

|           | QED             | DRD2            | HIV             | SARS            |
|-----------|-----------------|-----------------|-----------------|-----------------|
| LaMCTS*** | 0.914 +/- 0.002 | 0.323 +/- 0.016 | 0.406 +/- 0.019 | 0.452 +/- 0.010 |
| PlaLaM    | 0.916 +/- 0.001 | 0.648 +/- 0.026 | 0.588 +/- 0.020 | 0.570 +/- 0.018 |


&nbsp;

&nbsp;


\*While LaMCTS optimizes a *policy* rather than a *trajectory* in their original paper, it’s easily applicable to our path planning tasks as well. However, the original version of LaMCTS using TuRBO as the inner leaf solver is very slow, so we run LaMCTS on all of our tasks using CMA-ES as the inner leaf solver -- the same as PlaLaM. We also emphasize that both our PlaLaM and LaMCTS are meta-algorithms that can operate on top of any inner leaf solver.

Nevertheless, we run small-scale experiments (32 trial seeds instead of 256, hence larger std) on our MiniWorld environments for LaMCTS using the TuRBO solver as in their original paper:

|                | MazeS3       | FourRooms    | SelectObj    |
|----------------|--------------|--------------|--------------|
| LaMCTS(TuRBO)  | 21.9 +/- 7.3 | 0.0 +/- 0.0  | 0.0 +/- 0.0  |
| LaMCTS(CMA-ES) | 23.4 +/- 2.6 | 20.3 +/- 2.5 | 0.8 +/- 0.6  |
| PlaLaM(CMA-ES) | 57.0 +/- 3.1 | 89.1 +/- 2.0 | 45.3 +/- 3.1 |


PlaLaM outperforms the TuRBO version of LaMCTS here as well (previous results using CMA-ES inner solver reproduced for convenience). We hypothesize the poor TuRBO results are partially due to our smaller sample budget (2000 to 4000) compared to most experiments in the original LaMCTS paper (up to 40000 on complex envs); the TuRBO version draws many more samples per inner optimization compared to our CMA-ES version, leading to less tree exploration later.

**MiniGrid numbers differ slightly from the original draft, following reviewer vLZD’s comment about keeping latent spaces consistent.

***For molecular design, it’s not a fair comparison with the latent space removed, since the latent space is pretrained in this task only; all methods struggle to generate valid molecules of reasonable length without it. So we run LaMCTS with a latent space in this task only (the difference with PlaLaM is then just using the mean vs max observed reward).

---

### Decision · Program_Chairs · 2021-09-27

**Decision:**

Accept (Poster)

**Comment:**

The paper focuses on a black-box search approach to planning problems. The paper provides a theoretical analysis of an existing method in this space - La-MCTS - and suggests a modification of that method grounded in their analysis, resulting in a new approach called PlaLam. This analysis was valued by all reviewers and considered technically sound. And of course, planning problems are a relevant problem to the machine learning community.

Although the authors did compare their method to various method on a variety of planning problems, initially, it was unclear how the performance of the proposed 'PlaLam' modification compares to the original La-MCTS. This was brought up in review and was addressed by the authors in their reply. Other, minor, concerns of the reviewers were also satisfactorily addressed by the authors.

The paper was considered to be well written.